# ADDING CONDITIONAL CONTROL TO DIFFUSION MODELS WITH REINFORCEMENT LEARNING

**Yulai Zhao**[*][†]
Princeton University
yulaiz@princeton.edu

**Masatoshi Uehara**[*]
Genentech
uehara.masatoshi@gene.com

**Gabriele Scalia**
Genentech
scaliag@gene.com

**Sunyuan Kung**
Princeton University
kung@princeton.edu

**Tommaso Biancalani**
Genentech
biancalt@gene.com

**Sergey Levine**[‡]
University of California, Berkeley
sergey.levine@berkeley.edu

**Ehsan Hajiramezanali**[‡]
Genentech
hajiramm@gene.com

## ABSTRACT

Diffusion models are powerful generative models that allow for precise control over the characteristics of the generated samples. While these diffusion models trained on large datasets have achieved success, there is often a need to introduce additional controls in downstream fine-tuning processes, treating these powerful models as pre-trained diffusion models. This work presents a novel method based on reinforcement learning (RL) to add such controls using an offline dataset comprising inputs and labels. We formulate this task as an RL problem, with the classifier learned from the offline dataset and the KL divergence against pre-trained models serving as the reward functions. Our method, **CTRL** (**C**onditioning pre-**T**rained diffusion models with **R**einforcement **L**earning), produces soft-optimal policies that maximize the abovementioned reward functions. We formally demonstrate that our method enables sampling from the conditional distribution with additional controls during inference. Our RL-based approach offers several advantages over existing methods. Compared to classifier-free guidance, it improves sample efficiency and can greatly simplify dataset construction by leveraging conditional independence between the inputs and additional controls. Additionally, unlike classifier guidance, it eliminates the need to train classifiers from intermediate states to additional controls. The code is available at https://github.com/zhaoyl18/CTRL.

## 1 INTRODUCTION

Diffusion models have emerged as effective generative models for capturing intricate distributions (Sohl-Dickstein et al., 2015; Ho et al., 2020). Their capabilities are further enhanced by building conditional diffusion models $p(x|c)$. For instance, in text-to-image generative models like DALL-E (Ramesh et al., 2021) and Stable Diffusion (Rombach et al., 2022), $c \in \mathcal{C}$ is a prompt, and $x \in \mathcal{X}$ is the image generated according to this prompt. While diffusion models trained on extensive datasets have shown remarkable success, additional controls often need to be incorporated during downstream fine-tuning when treating these powerful models as pre-trained diffusion models.

In this work, our goal is to incorporate new conditional controls into pre-trained diffusion models. Specifically, given access to a large pre-trained model capable of modeling $p(x|c)$ trained on extensive datasets, we aim to condition it on an additional random variable $y \in \mathcal{Y}$, thereby creating a generative model $p(x|c, y)$. To accomplish this, we utilize the pre-trained model and an offline dataset consisting

---

[*]Equal contribution.
[†]This work is done during an internship at Genentech.
[‡]Corresponding authors.

Table 1: Comparison with existing approaches. Unlike classifier guidance and its variations, our method involves directly fine-tuning pre-trained models. We avoid the need to learn a mapping from $x_t \to y$ or rely on heuristic approximations. Additionally, while classifier-free guidance always demands triplets $\{c, x, y\}$, our approach can leverage conditional independence ($y \perp c|x$) and only necessitate pairs $\{x, y\}$, simplifying the construction of the offline dataset.

| Methods | Fine-tuning | Need to learn $x_t \to y$ | Leveraging conditional independence |
|---|---|---|---|
| Classifier guidance (Dhariwal and Nichol, 2021) | No | Yes | Yes |
| Reconstruction guidance (e.g. Ho et al. (2022); Chung et al. (2022); Han et al. (2022)) | No | No | Yes |
| Classifier-free guidance (Ho and Salimans, 2022) | Yes | No | No |
| **CTRL** (Ours) | Yes | No | Yes |

of triplets $\{c, x, y\}$. This scenario is important, as highlighted in the existing literature on computer vision (e.g., Zhang et al. (2023)), because it enables the extension of generative capabilities with new conditional variables without requiring retraining from scratch. Currently, classifier-free guidance (Ho et al., 2020) is a prevailing approach for incorporating conditional controls into diffusion models, and it has proven successful in computer vision (Zhang et al., 2023; Zhao et al., 2024). However, its effectiveness may not extend well to other challenging problems, especially when large offline datasets are unavailable. Indeed, the success of training conditional diffusion models via classifier-free guidance heavily relies on such datasets (Brooks et al., 2023), which are often impractical to obtain. In these scenarios, this method tends to struggle.

In our work, we present a new approach for adding new conditional controls via reinforcement learning (RL) to further improve sample efficiency. Inspired by recent progress in RL-based fine-tuning (Black et al., 2023; Fan et al., 2023; Uehara et al., 2024), we frame the conditional generation as an RL problem within a Markov Decision Process (MDP). In this formulation, the reward, which we want to maximize, is the (conditional) log-likelihood function $\log p(y|x, c)$, and the policy, conditioned on $(c, y)$, corresponds to the denoising process at each time step in a diffusion model. We formally demonstrate that, by executing the soft-optimal policy, which maximizes the reward $\log p(y|x, c)$ with KL penalty against the pre-trained model, we can sample from the target conditional distribution $p(x|c, y)$ during inference. Hence, our proposed algorithm, **CTRL** (**C**onditioning pre-**T**rained diffusion models with **R**einforcement **L**earning) consists of three main steps: (1) learning a classifier $\log p(y|x, c)$ (which will serve as our reward function in the MDP) from the offline dataset, (2) constructing an augmented diffusion model by adding (trainable) parameters to the pre-trained model in order to accommodate an additional label $y$, and (3) learning soft-optimal policy within the aforementioned MDP during fine-tuning. Our approach is novel as it significantly diverges from classifier-free guidance and distinguishes itself from existing RL-based fine-tuning methods by integrating an augmented model in the fine-tuning process to support additional controls.

Our novel RL-based approach offers several advantages over existing methods for adding additional controls. Firstly, in contrast to classifier-free guidance, which uses offline data to directly model $p(x|y, c)$, our method leverages offline data to model the simpler distribution $p(y|x, c)$, improving sample efficiency (in typical scenarios where $y$ is lower-dimensional than $x$). Secondly, in typical scenarios where the additional label $y$ depends solely on $x$ (e.g., the compressibility of an image depends only on the image, not the prompt), our fine-tuning method only requires pairs $\{x, y\}$, whereas classifier-free guidance still necessitates triplets $\{c, x, y\}$ from the offline dataset. This is because the reward function simplifies to $\log p(y|x)$ due to the conditional independence $y \perp c|x$, which gives $\log p(y|x, c) = \log p(y|x)$. Furthermore, when the goal is to simultaneously add conditioning controls on two labels, $y_1$ and $y_2$, and both labels only depend on $x$, our method requires only pairs $\{x, y_1\}$ and $\{x, y_2\}$. In contrast, classifier-free guidance requires quadruples $\{c, x, y_1, y_2\}$. Therefore, in this manner, **CTRL** can also leverage the *compositional nature* of the mapping between inputs and additional labels.

Our contributions can be summarized as follows. We propose an RL-based fine-tuning approach for conditioning pre-trained diffusion models on additional labels. In comparison to classifier-free

guidance, our method uses the offline dataset in a sample-efficient manner and enables leveraging the conditional independence assumption, which significantly simplifies the construction of the offline dataset. Additionally, we establish a close connection to classifier guidance (Dhariwal and Nichol, 2021; Song et al., 2020), showing that it provides an alternative method for obtaining the aforementioned soft-optimal policies (in ideal cases where there are no statistical/model-misspecification errors in the algorithms). Despite this connection, our algorithm addresses common challenges in classifier guidance, such as the need to learn classifiers at multiple noise scales in standard classifier guidance and the use of fundamental approximations in some variants to avoid learning these noisy classifiers (Chung et al., 2022; Song et al., 2022). Experimentally, we validate the superiority of **CTRL** over baselines in both single-task and multi-task conditional image generation, such as generating highly aesthetic yet compressible images, where existing methods often struggle. Table 1 summarizes the main features of the proposed algorithm compared to existing methods.

## 2 RELATED WORKS

**Classfier guidance.** Dhariwal and Nichol (2021); Song et al. (2020) introduced classifier guidance, a method that entails training a classifier and incorporating its gradients to guide inference (while freezing pre-trained models). However, a notable drawback of this technique lies in the classifier's accuracy in predicting $y$ from intermediate $x_t$, resulting in cumulative errors during the diffusion process. To mitigate this issue, several studies propose methods to circumvent it through reconstruction, referred to as **reconstruction guidance** in this work. Specifically, they employ certain approximations that map intermediate states $x_t$ back to the original input space $x_0$, allowing the classifier to be learned solely from $x_0$ to $y$ (Ho et al., 2022; Han et al., 2022; Chung et al., 2022; Finzi et al., 2023; Bansal et al., 2023). In contrast to these works, our approach focuses on fine-tuning the diffusion model itself rather than relying on an inference-time technique. While the strict comparison between model fine-tuning and inference-time techniques is not feasible, we theoretically elucidate the distinctions and connections of our approach with classifier guidance in Section 5.1.

**Classfier-free guidance.** Classifier-free guidance (Ho and Salimans, 2022) is a method that directly conditions the generative process on both data and context, bypassing the need for explicit classifiers. This methodology has been widely and effectively applied, for example, in text-to-image models (Nichol et al., 2021; Saharia et al., 2022; Rombach et al., 2022). While the original research does not explore classifier-free guidance within the scope of fine-tuning pre-trained diffusion models, several subsequent studies address fine-tuning scenarios Zhang et al. (2023); Xie et al. (2023). As elucidated in Section 5.2, compared to classifier-free guidance, our approach can improve sample efficiency and leverage conditional independence to facilitate offline dataset construction.

**Fine-tuning via RL.** Several previous studies have addressed the fine-tuning of diffusion models by optimizing relevant reward functions. Methodologically, these approaches encompass supervised learning (Lee et al., 2023; Wu et al., 2023), reinforcement learning (Black et al., 2023; Fan et al., 2023; Uehara et al., 2024), and control-based techniques (Clark et al., 2023; Xu et al., 2023; Prabhudesai et al., 2023; Uehara et al., 2024). While our proposal draws inspiration from these works, our objective for fine-tuning is to tackle a distinct goal: incorporating *additional* controls. To achieve this, unlike previous approaches, we employ policies with augmented parameters rather than merely fine-tuning pre-trained models without adding any new parameters.

**Remark 1.** *A concurrent study (Denker et al., 2025) presents a similar key theorem demonstrating that conditioning can be framed as reinforcement learning (RL) problems. However, while their experiments rely on denoising score-matching objectives, our work empirically validates the effectiveness of RL objectives.*

## 3 PRELIMINARIES

In this section, we introduce the problem setting, review the existing methods addressing this problem, and discuss their disadvantages.

### 3.1 GOAL: CONDITIONING WITH ADDITIONAL LABELS USING OFFLINE DATA

We first define our main setting and main objective. Throughout this paper, we use $\mathcal{Y}$ and $\mathcal{C}$ to represent condition spaces and $\mathcal{X}$ to denote the (Euclidean) sample space. Given the pre-trained

model, which enables us to sample from $p^{\mathrm{pre}}(x|c) : \mathcal{C} \to \Delta(\mathcal{X})$, our goal is to add new conditional controls $y \in \mathcal{Y}$ such that we can sample from $p(x|c, y)$.

**Pre-trained model and offline dataset.** A (continuous-time) pre-trained conditional diffusion model is characterized by the following SDE[1], where $f^{\mathrm{pre}} : [0, T] \times \mathcal{C} \times \mathcal{X} \to \mathbb{R}^d$ is a model with parameter $\theta$:

$$dx_t = f^{\mathrm{pre}}(t, c, x_t; \theta^{\mathrm{pre}})dt + \sigma(t)dw_t, \quad x_0 = x_{\mathrm{ini}}, \tag{1}$$

In training diffusion models, the parameter $\theta^{\mathrm{pre}}$ is derived by optimizing a specific loss function on large datasets[2]. We refer interested readers to Appendix A for more details on constructing these loss functions. Using the pre-trained model and following the above SDE (1) from $0$ to $T$, we can sample from $p^{\mathrm{pre}}(\cdot|c)$ for any condition $c \in \mathcal{C}$.

To add additional control to a pre-trained model, as in many recent works (Dhariwal and Nichol, 2021; Bansal et al., 2023; Epstein et al., 2023), we assume access to offline data: $\mathcal{D} = \{c^{(i)}, x^{(i)}, y^{(i)}\}_{i=1}^n \in \mathcal{C} \times \mathcal{X} \times \mathcal{Y}$. We denote the conditional distribution of $y$ given $x$ and $c$ by $p^\diamond(y|x, c)$.

**Target distribution.** Using the pre-trained model and the offline dataset, our goal is to obtain a diffusion model such that we can sample from a distribution over $\mathcal{C} \times \mathcal{Y} \to \Delta(\mathcal{X})$ as below:

$$p_\gamma(\cdot|c, y) := \frac{\{p^\diamond(y|\cdot, c)\}^\gamma p^{\mathrm{pre}}(\cdot|c)}{\int \{p^\diamond(y|x, c)\}^\gamma p^{\mathrm{pre}}(x|c)\mu(dx)}, \tag{2}$$

where $\gamma \in \mathbb{R}^+$ denotes the strength of additional guidance and $\mu$ is the Lebsgue measure.

Such target distribution is extensively explored in the literature on classifier guidance and classifier-free guidance (Dhariwal and Nichol, 2021; Ho and Salimans, 2022; Nichol et al., 2021; Saharia et al., 2022; Rombach et al., 2022). Specifically, when $\gamma = 1$, this distribution corresponds to the standard conditional distribution $p(x|c, y)$, which is a fundamental objective of many conditional generative models (Dhariwal and Nichol, 2021; Ho and Salimans, 2022). Moreover, for a general $\gamma$, $p_\gamma$ can be formulated via the following optimization problem:

$$p_\gamma(\cdot|c, y) = \underset{q:\mathcal{C} \times \mathcal{Y} \to \Delta(\mathcal{X})}{\mathrm{argmin}} \mathbb{E}_{x \sim q(\cdot|c,y)}[-\gamma \log p^\diamond(y|x, c)] + \mathrm{KL}(q(\cdot|c, y)\|p^{\mathrm{pre}}(\cdot|c)).$$

This relation is clear as the objective function equals $\mathrm{KL}(q(\cdot|c, y)\|p_\gamma(\cdot|c, y))$ up to a constant.

**Goal.** As discussed, the primary goal of this research is to train a generative model capable of simulating $p_\gamma(\cdot|c, y)$. To achieve this, we introduce the following SDE:

$$dx_t = g(t, c, y, x_t)\,dt + \sigma(t)\,dw_t, \quad x_0 = x_{\mathrm{ini}}, \tag{3}$$

where $g : [0, T] \times \mathcal{C} \times \mathcal{Y} \times \mathcal{X} \to \mathbb{R}^d$ is an augmented model to add additional controls into pre-trained models. The primary challenge involves leveraging both offline data and pre-trained model weights to train the term $g$, ensuring that the marginal distribution of $x_T$ induced by the SDE (3) accurately approximates $p_\gamma$.

**Notation.** Let the space of trajectories $x_{0:T}$ be $\mathcal{K}$. Conditional on $c$ and $y$, we denote the measure induced by the SDE (3) over $\mathcal{K}$ by $\mathbb{P}^g(\cdot|c, y)$. Similarly, we use $\mathbb{P}_t^g(\cdot|c, y)$ and $p_t^g(\cdot|c, y)$ to represent the marginal distribution of $x_t$ and density $d\mathbb{P}_t^g(\tau|c, y)/d\mu$.

## 3.2 Existing Methods

In this subsection, we describe two existing methods that are applicable in our context to achieve the aforementioned goal.

### 3.2.1 Classfier-Free Guidance

Recent works have studied fine-tuning pre-trained models with classifier-free guidance (Brooks et al., 2023; Zhang et al., 2023; Xie et al., 2023). These methods introduce an augmented model $g$ as described in (3), where the weights are initialized from the pre-trained model. Fine-tuning is via minimizing classifier-free guidance loss on the offline dataset. While successful in many applications, these methods may struggle in scenarios where offline datasets for new conditions are limited (Huang et al., 2021; Yellapragada et al., 2024; Giannone et al., 2024).

---

[1]For simplicity, we consider the case where the initial distribution is a Dirac delta, as in bridge matching. The extension of our proposal for stochastic distributions remains straightforward (Uehara et al., 2024).

[2]For notational simplicity, throughout this work, we would often drop $\theta^{\mathrm{pre}}$.

### 3.2.2 CLASSFIER GUIDANCE

Classifier guidance (Dhariwal and Nichol, 2021; Song et al., 2020) is based on the following result.

**Lemma 1** (Doob's h-transforms (Rogers and Williams, 2000)). *For any $c \in \mathcal{C}$ and $y \in \mathcal{Y}$, by evolving according to the following SDE from 0 to $T$:*

$$dx_t = \{f^{\mathrm{pre}}(t, c, x_t) + \sigma^2(t) \underbrace{\nabla_{x_t} \log \mathbb{E}_{\substack{x_{t:T} \sim \mathbb{P}^{\mathrm{pre}}(\cdot|x_t, c) \\ y' \sim p^{\diamond}(\cdot|x_T, c)}}[\mathrm{I}(y = y')|x_T, c]}_{\text{Additional Drift}:=\nabla_{x_t} \log p(y|x_t, c)}\}dt + \sigma(t)dw_t, \quad (4)$$

*the marginal distribution of $x_T$, i.e., $p(x_T|c, y)$, is equal to the target distribution $p_{\gamma=1}(\cdot|c, y)$ (2). Here, $\mathbb{P}^{\mathrm{pre}}$ denotes the distribution induced by the pre-trained diffusion model (1).*

This lemma suggests that to simulate the target distribution (2), we only need to construct SDE (4). However, practical issues arise: first, training classifier $p(y|x_t, c)$ requires extensive data at each timestep, which is cumbersome with large pre-trained models. Furthermore, accumulated inaccuracies in drift estimates may lead to poor performance (Li and van der Schaar, 2023).

**Reconstruction guidance.** To mitigate these issues, several studies propose to approximate $p(y|x_t, c)$ directly via reconstruction (Ho et al., 2022; Han et al., 2022; Chung et al., 2022; Guo et al., 2024)[3], specifically by $p(y|x_t, c) = \int p^{\diamond}(y|x_T, c)p(x_T|x_t, c)dx_T \approx p(y|\hat{x}_T(x_t, c), c)$, where $\hat{x}_T(x_t, c)$ is the (expected) denoised sample given $x_t, c$, i.e., $\hat{x}_T(x_t, c) = \mathbb{E}[x_T|x_t, c]$. Given such an approximation, we only need to learn $p^{\diamond}(y|x_T, c)$ from data. However, this approximation may become imprecise when $\mathbb{P}(x_T|x_t, c)$ is noisy or is difficult to estimate reliably (Chung et al., 2022).

## 4 CONDITIONING PRE-TRAINED DIFFUSION MODELS WITH RL

This section provides details on how our method solves the aforementioned goal with methodological motivations. We begin with a key observation: the conditioning problem can be effectively conceptualized as an RL problem.[4] Building upon this insight, we illustrate our main algorithm.

### 4.1 CONDITIONING AS RL

Recall that our objective is to learn a drift term $g$ in (3) so that the induced marginal distribution at $T$ (i.e., $p_T^g$) closely matches our target distribution $p_\gamma$. To achieve this, we first formulate the problem via the following minimization:

$$\operatorname*{argmin}_g \mathrm{KL}(p_T^g(\cdot|c, y) \| p_\gamma(\cdot|c, y)).$$

With some algebra, we can show that the above optimization problem is equivalent to the following:

$$\operatorname*{argmin}_g \mathbb{E}_{x_{0:T} \sim \mathbb{P}^g(\cdot|c,y)} \left[ -\gamma \log p^{\diamond}(y|x, c) + \frac{1}{2} \int_0^T \frac{\|f^{\mathrm{pre}}(s, c, x_s) - g(s, c, y, x_s)\|^2}{\sigma^2(s)} ds \right].$$

Here, recall that $\mathbb{P}^g$ is the measure induced by SDE (3) with a drift coefficient $g$. Based on this observation, we derive the following theorem.

**Theorem 1** (Conditioning as RL). *Consider the following RL problem:*

$$g^\star := \operatorname*{argmax}_g \mathbb{E}_{\substack{(c,y) \sim \Pi(c,y) \\ x_{0:T} \sim \mathbb{P}^g(\cdot|c,y)}} \left[ \gamma \log p^{\diamond}(y|x_T, c) - \frac{1}{2} \int_0^T \frac{\|f^{\mathrm{pre}}(s, c, x_s) - g(s, c, y, x_s)\|^2}{\sigma^2(s)} ds \right], \quad (5)$$

*where $\Pi \in \Delta(\mathcal{C} \times \mathcal{Y})$. Significantly, the marginal distribution $p_T^{g^\star}$ matches our target distribution:*

$$\forall (c, y) \in \mathrm{Supp}(\Pi); \ p_T^{g^\star}(\cdot|c, y) = p^\gamma(\cdot|c, y).$$

Proofs are deferred to Appendix D. This theorem demonstrates that, after obtaining the optimal drift $g^\star$ by solving the RL problem in (5), we can sample from the target distribution $p^\gamma(\cdot|c, y)$ by following SDE (3) from time 0 to $T$. In the next section, we explain how to solve (5) in practice.

### 4.2 ALGORITHM

Theoretically inspired by Theorem 1, we introduce Algorithm 1. It consists of three steps.

---

[3]We categorize them as reconstruction guidance methods for simplicity. We note that there are many variants.

[4]For more details of the RL formulation, such as state space, action space, and transition function, please refer to (Uehara et al., 2024).

---

**Algorithm 1** **C**onditioning pre-**T**rained diffusion models with **R**einforcement **L**earning (**CTRL**)

---

1: **Input**: Pre-trained model with a drift coefficient $f^{\mathrm{pre}}$, Offline data $\mathcal{D} = \{c^{(i)}, x^{(i)}, y^{(i)}\}$, Exploratory distribution $\Pi \in \Delta(\mathcal{C} \times \mathcal{Y})$
2: Construct an augmented model $g(t, c, y, x; \psi)$.
3: Train a classifier $\hat{p}(y|x, c)$ to approximate $p^{\diamond}(y|x, c)$ from the offline data $\mathcal{D}$
4: Fine-tune the diffusion model by solving the following RL problem (e.g. using Algorithm 2):

$$\hat{\psi} = \underset{\psi}{\operatorname{argmax}} \, \mathbb{E}_{\substack{(c,y) \sim \Pi(c,y) \\ x_{0:T} \sim \mathbb{P}^g(\cdot|c,y;\psi)}} \left[ \gamma \log \hat{p}(y|x_T, c) - \frac{1}{2} \int_0^T \frac{\|f^{\mathrm{pre}}(s, c, x_s) - g(s, c, y, x_s; \psi)\|^2}{\sigma^2(s)} ds \right]$$

where $\mathbb{P}^g(\cdot|c, y; \psi)$ is an distribution induced by the SDE with a parameter $\psi$.
5: **Output**: $dx_t = g(t, c, y, x_t; \hat{\psi})dt + \sigma(t)dw_t$

---

**Step 1: Constructing the augmented model (Line 2).** To add additional conditioning to the pretrained diffusion model, it is necessary to enhance the pre-trained model $f^{\mathrm{pre}}(t, c, x; \theta)$. We introduce an augmented model $g(t, c, y, x; \psi)$ with parameters $\psi = [\theta^{\top}, \phi^{\top}]^{\top}$, initialized at $\psi^{\mathrm{ini}} = [\theta^{\mathrm{pre}\top}, \mathbf{0}^{\top}]$. Here, $\psi$ is structured as a combination of the existing parameters $\theta$ and new parameters $\phi$.

Determining the specific architecture of the augmented model involves a tradeoff: adding more new parameters enhances expressiveness but raises computational costs. In scenarios where $\mathcal{Y}$ is discrete with cardinality $|\mathcal{Y}|$, the most straightforward solution is to instantiate $\phi$ with a simple linear embedding layer that maps each $y \in \mathcal{Y}$ to its corresponding embedding. These embeddings are then added to every intermediate output in the diffusion SDE (i.e., $x_t$ in (3)). This method preserves the original structure to the fullest extent while ensuring that all pre-trained weights are fully utilized. Experimentally, we observe that this lightweight modification leads to accurate conditional generations for complex conditioning tasks, as shown in Section 6.

**Step 2: Training a calibrated classifier with offline data (Line 3).** Using a function class $\mathcal{F} \subset [\mathcal{C} \times \mathcal{X} \to \Delta(\mathcal{Y})]$, we perform maximum likelihood estimation (MLE):

$$\hat{p}(\cdot|x, c) := \underset{r \in \mathcal{F}}{\operatorname{argmax}} \sum_{i=1}^{n} \log r(y^{(i)}|x^{(i)}, c^{(i)}). \tag{6}$$

For instance, when $\mathcal{Y}$ is discrete, this loss reduces to the standard cross-entropy loss. When $\mathcal{Y}$ is continuous, assuming Gaussian noise, it reduces to a regression loss.

**Step 3: Planning (Line 4).** Equipped with a classifier, we proceed to solve the RL problem (5), which constitutes the core of the proposed algorithm. As noted by Black et al. (2023); Fan et al. (2023), the diffusion model can be regarded as a special Markov Decision Process (MDP) with known transition dynamics. Thus, many types of off-the-shelf RL algorithms can be employed for planning. In this work, inspired by (Clark et al., 2023; Prabhudesai et al., 2023), we employ direct backpropagation, which requires differentiable. If a classifier is not differentiable, we recommend using PPO-based methods (Schulman et al., 2017; Black et al., 2023). Details are deferred to Appendix B.

Below, we make several remarks regarding implementing **CTRL** in practice.

**Remark 2** (Using classifier-free guidance to adjust guidance strength). *Throughout the fine-tuning process demonstrated in Algorithm 1, the guidance strength for the additional conditional control (i.e., $y$) is fixed at a specific $\gamma$ (see the target conditional distribution (2)). However, we note that during inference, this guidance strength $\gamma$ can be adjusted—either increased or decreased—using the classifier-free guidance technique. Details are deferred to Appendix C.*

**Remark 3** (Choice of exploratory distribution $\Pi$). *According to Theorem 1, it is desired to improve the coverage over $\mathcal{C} \times \mathcal{Y}$ during fine-tuning. For example, in practice, if $\mathcal{Y}$ only takes several discrete values, we can sample $y \in \mathcal{Y}$ uniformly from these values as done in Section 6.*

### 4.3 SOURCE OF ERRORS IN **CTRL**

We discuss the potential sources of error that **CTRL** may encounter, which will be useful for comparison with existing methods in the next section. Additional limitations, such as computational cost, memory complexity, and the choice of guidance strength $\gamma$, are discussed in Appendix F.

**Statistical error.** Statistical errors arise during the training of a classifier $\hat{p}(y|x, c)$ from offline data while learning $p^\diamond(y|x, c)$. A typical statistical error is given by:

$$\mathbb{E}_{(x,c)\sim l^{\text{off}}}[\|\hat{p}(\cdot|x, c) - p^\diamond(\cdot|x, c)\|_1^2] = O(\text{Cap}(\mathcal{F})/n), \qquad (7)$$

where $l^{\text{off}} \in \Delta(\mathcal{X} \times \mathcal{C})$ represents the distribution of offline data, and $\text{Cap}(\mathcal{F})$ denotes the size of the function class $\mathcal{F}$ (Wainwright, 2019).

**Model-misspecification error.** Model-misspecification errors may occur during the learning of the classifier $p^\diamond(y|x, c)$ and in the augmented model if it fails to capture the optimal drift $g^\star$.

**Optimization error.** Optimization errors may occur during both the classifier training step and the planning step.

## 5 Additional Comparisons with Existing Conditioning Methods

In this section, we further clarify the connections and comparisons between our algorithm and the existing methods.

### 5.1 Comparison to Classifier Guidance

We explore the advantages of **CTRL** over classifier guidance (Dhariwal and Nichol, 2021), while also offering theoretical insights that link the two approaches. Despite their distinct goals – classifier guidance is an inference-time technique, whereas our method fine-tunes an augmented diffusion model, there is a deep theoretical connection. This link is highlighted by our derivation of the analytical expression for the optimal drift in the RL problem (5) as below.

**Lemma 2** (Bridging RL-based conditioning with classifier guidance). *The optimal drift term $g^\star$ for RL problem* (5) *has the following explicit solution:*

$$g^\star(t, c, y, x_t) = f^{\text{pre}}(t, c, x_t) + \sigma^2(t)\nabla_{x_t} \log \mathbb{E}_{\mathbb{P}^{\text{pre}}(\cdot|x_t, c)} [(p^\diamond(y|x_T, c))^\gamma|x_t, c], \quad \forall t \in [0, T]$$

The proof of Lemma 2 is deferred to Appendix D.3. This lemma indicates that when $\gamma = 1$, the optimal drift $g^*$ corresponds to the drift term obtained from Doob's h-transform (i.e., Lemma 1), which is a precise used formula in classifier guidance. Despite the link to classifier guidance through Lemma 2, our algorithm is fundamentally different. Classifier guidance requires learning a predictor from $x_t$ to $y$ for every $t \in T$, leading to accumulated inaccuracies. In contrast, our algorithm directly solves the RL problem (5), avoiding the need for such predictors.

In Section 3.2.2, we explore **reconstruction guidance** methods that also aim to circumvent predicting $y$ from $x_t$. These methods propose first mapping $x_t$ to a denoised estimate $\hat{x}_T(x_t)$ and using this for further computations. However, this approximation can be imprecise, especially over longer time horizons. As shown in (Chung et al., 2022, Theorem 1), inherent approximation errors persist even without statistical, model-misspecification, or optimization errors. In contrast, our algorithm avoids such approximation errors.

### 5.2 Comparison to Classifier-Free Guidance

We first show how **CTRL** leverages *conditional independence* to ease implementation, a feature absent in classifier-free guidance. Finally, we discuss the improvements regarding sample (statistical) efficiency.

#### 5.2.1 Leveraging Conditional Independence, Compositionally via **CTRL**

We discuss two scenarios where our method outperforms the classifier-free approach by exploiting the conditional independence between inputs and additional controls.

**Example 1** (Scenario $Y \perp C|X$). *If a new condition $Y$ is conditionally independent of an existing condition $C$ given $X$, meaning $p^\diamond(y|x, c) = p(y|x)$. This allows **CTRL** to operate efficiently with just $(x, y)$ pairs, avoiding the need for $(c, x, y)$ triplets in the offline dataset.*

This scenario is common in practice. For example, when using the Stable Diffusion pre-trained model (Rombach et al., 2022), where $X$ is an image and $C$ is a text prompt, we may also want to condition the generations on $Y$, such as score functions like compressibility, aesthetic score, or color (Black et al., 2023). These scores depend solely on the image and are independent of the prompt, meaning $Y \perp C|X$. We further explore this scenario in our experimental analysis in Section 6.1.

**Multi-task conditional generation.** Multi-task conditional generation is a significant challenge, requiring the integration of multiple controls into pre-trained models. In the following example, we show how our method can be extended to handle this.

**Example 2** (Scenario $Y_1 \perp Y_2 | X, C$). *If two conditions, $Y_1$ and $Y_2$, exhibit conditional independence given $X$ and $C$, such that $\log p(y_1, y_2 | x, c) = \log p(y_1 | x, c) + \log p(y_2 | x, c)$, the two classifiers can be trained separately using $(c, x, y_1)$ and $(c, x, y_2)$ triplets. Furthermore, if $Y_1$ and $Y_2$ are also independent of $C$ given $X$ (as in Example 1), the classifiers can be trained solely with $(x, y_1)$ and $(x, y_2)$ pairs, significantly simplifying dataset construction.*

This scenario is also common in practice. For instance, with the Stable Diffusion pre-trained model, where $X$ is an image and $C$ is a text prompt, additional attributes like $Y_1$ (compressibility) and $Y_2$ (color) depend only on the image, not the prompt. Thus, we can leverage the conditional independence of $Y_1$ and $Y_2$ from $C$ given $X$ to simplify the implementation of **CTRL**. The effectiveness of **CTRL** in this context is further validated experimentally in Section 6.2.

**Can classifier-free guidance leverage conditional independence?** The applicability of conditional independence in classifier-free guidance, which directly models $p_\gamma(\cdot | c, y)$, is uncertain. For instance, when $Y \perp C | X$ as in Example 1, our method only requires $(x, y)$ pairs, while classifier-free guidance typically needs $(c, x, y)$ triplets. when $Y_1 \perp Y_2 | C, X$ as in Example 2, our approach utilizes triplets $(c, x, y_1)$ and $(c, x, y_2)$. However, as far as we are concerned, quadruples $(c, x, y_1, y_2)$ are necessary for classifier-free guidance, and acquiring such data at scale could pose a bottleneck.

### 5.2.2 STATISTICAL EFFICIENCY

We present the rationale for our approach being more sample-efficient than classifier-free guidance. Most importantly, we leverage a pre-trained model to sample from $p^{\mathrm{pre}}(x|c)$, which is already trained on large datasets. This allows us to focus only on learning the classifier $p^\diamond(y|x, c)$ from offline data. As a result, any statistical errors from the offline data affect only the classifier learning step (6). In contrast, classifier-free guidance attempts to model the entire distribution $p_\gamma(\cdot | c, y)$ directly from offline data. Therefore, our method is more sample-efficient by learning only the necessary components from the offline data.

## 6 EXPERIMENTS

We compare **CTRL** with five baselines: (1) **Reconstruction Guidance**. It attempts to alleviate the approximation error of classifier guidance via reconstruction. (2) **Classifier-Free** guidance (Ho and Salimans, 2022)[5]. (3) **SMC** (Sequential Monte Carlo). Recent works (Wu et al., 2024; Phillips et al., 2024) leverage resampling techniques to approximate distributions in diffusion models across a batch of samples (i.e., particle filtering). This method is training-free. (4) **SVDD** (Li et al., 2024). It is a decoding-based method that iteratively selects preferable samples during each diffusion step based on reward signals. For our conditioning task, we use a trained classifier as the reward function. This method is also training-free and operates at inference time. (5) **MPGD** (He et al., 2024). This method optimizes the predicted clean data $x_0$ during inference based on a manifold hypothesis. While such refinement incurs additional computational costs during inference, it does not fine-tune diffusion model weights, remaining a training-free approach. For more detailed information on each experiment, such as dataset, architecture, and baselines, please refer to Appendix E.

**Experimental setup.** For image experiments (Section 6.1, Section 6.2), we use Stable Diffusion v1.5 (Rombach et al., 2022) as the pre-trained model $p^{\mathrm{pre}}(x|c)$, here $c$ is a text prompt (e.g., "cat" or "dog") and $x$ is the corresponding image. For the additional control $y$, we validate compressibilities and aesthetic scores.

### 6.1 IMAGE: CONDITIONAL GENERATION ON COMPRESSIBILITY

We start by conditioning generations on their file sizes, specifically focusing on **compressibility** [6]. Denoting compressibility as **CP**, we define 4 compressibility labels as follows: $Y = 0 : \mathbf{CP} <$

---

[5]Implementing the **Classifier-Free** baseline in our setting would require using the pre-trained diffusion model to augment $x$ on certain $c$. Please refer to Appendix E.1 for details.

[6]Unlike standard tasks in classifier/reconstruction guidance (Chung et al., 2022), this score is non-differentiable w.r.t. images. This score function is only dependent on the image itself.

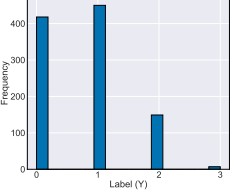

(a) Histogram.

| | Accuracy ↑ | Macro F1 score ↑ |
|---|---|---|
| **Classifier-Free** | 0.33 ± 0.04 | 0.28 |
| **Reconstruction Guidance** | 0.45 ± 0.04 | 0.45 |
| **SMC** | 0.27 ± 0.04 | 0.22 |
| **MPGD** | 0.70 ± 0.04 | 0.72 |
| **SVDD** | 0.78 ± 0.03 | 0.78 |
| **CTRL** | **1.0 ± 0.0** | **1.0** |

(b) Evaluation.

| | BRISQUE ↓ | CLIP Score ↑ |
|---|---|---|
| **CTRL** | 30.8 ± 1.5 | 26.8 ± 0.2 |
| **Classifier-Free** | 33.2 ± 14.5 | 25.7 ± 1.0 |
| **Reconstruction Guidance** | 90.2 ± 10.1 | 22.8 ± 0.6 |

(c) Image quality.

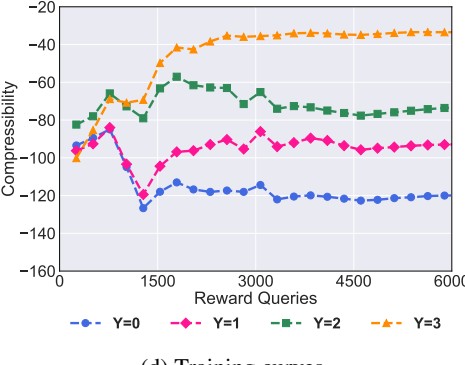

(d) Training curves.

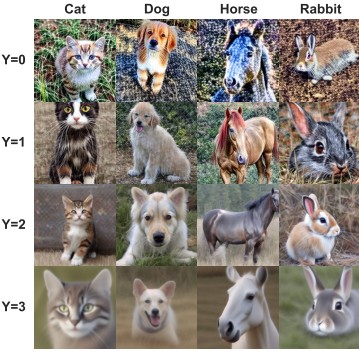

(e) Images generated by **CTRL**.

Figure 1: Results for conditioning on compressibility. Figure a plots the histogram of samples generated by the pre-trained diffusion model. Figure d shows the mean compressibility curves during fine-tuning with four distinct lines representing each condition. It is evident that **CTRL** effectively aligns the generated samples with their target compressibility levels via fine-tuning. Table b, c provide evaluation metrics, and Figure e shows images generated by a **single** model fine-tuned with **CTRL**.

$-110.0$; $Y = 1 : -110.0 \leq \mathbf{CP} < -85.0$; $Y = 2 : -85.0 \leq \mathbf{CP} < -60.0$; $Y = 3 : \mathbf{CP} \geq -60.0$. Particularly, as depicted in Figure 1a, generating samples conditioned on $Y = 3$ is challenging due to the infrequent occurrence of such samples from the pre-trained model.

**Results.** We evaluate performance across four compressibility levels using the following steps: (1) generating samples conditioned on each $Y \in [0, 1, 2, 3]$; (2) verifying alignment between the generated samples and their conditions; and (3) calculating classification accuracy and macro F1 score. Table 1b presents evaluation statistics. Our results show that **CTRL** accurately generates samples for each condition, including the rare $Y = 3$ case from the pre-trained model (see Figure 1a), notably outperforming the baselines. Figure 1e showcases diverse images with correct compressibility levels for various prompts. Additional visualizations are available in Appendix E.5.

In Table 1c, we provide additional evaluation metrics for baseline methods and **CTRL**, specifically BRISQUE (Mittal et al., 2012) (lower values indicate better image quality) and CLIPScore (Zhengwentai, 2023) (higher values reflect better text-image alignment). It is evident that **CTRL** achieves better image quality while achieving the highest alignment score.

### 6.2 IMAGE: MULTI-TASK CONDITIONAL GENERATION

A more challenging setting involves multi-task conditional generation. In this experiment, in addition to compressibility, we *simultaneously* aim to condition the generations on their aesthetic pleasingness. Following prior research (Black et al., 2023; Fan et al., 2023; Uehara et al., 2024), we employ an aesthetic scorer implemented as a linear MLP on top of the CLIP embeddings (Radford et al., 2021), which is trained on more than $400k$ human evaluations.

In this experiment, by leveraging conditional independence of $Y_1$ and $Y_1$ given $X$ (see Example 2), we aim to fine-tune the diffusion model to generate samples with *compositional* conditions. Specifically, denoting compressibility as $\mathbf{CP}$ and aesthetic score as $\mathbf{AS}$, we define four compositional conditions as follows: $Y = 0 : \mathbf{AS} < 5.7, \mathbf{CP} < -70$; $Y = 1 : \mathbf{AS} < 5.7, \mathbf{CP} \geq -70$; $Y = 2 : \mathbf{AS} \geq 5.7, \mathbf{CP} < -70$; $Y = 3 : \mathbf{AS} \geq 5.7, \mathbf{CP} \geq -70$. Particularly, as depicted in Figure 2a,

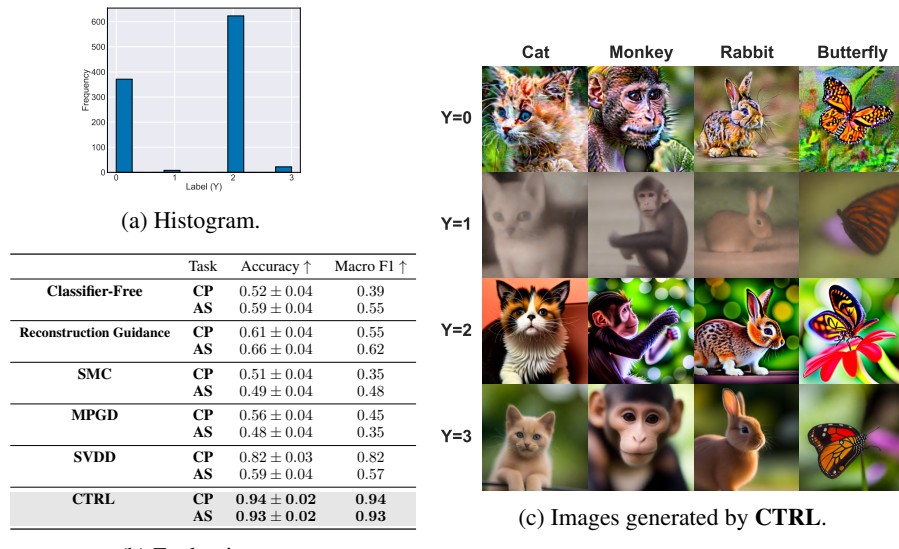

(a) Histogram.

| | Task | Accuracy ↑ | Macro F1 ↑ |
|---|---|---|---|
| **Classifier-Free** | CP | $0.52 \pm 0.04$ | 0.39 |
| | AS | $0.59 \pm 0.04$ | 0.55 |
| **Reconstruction Guidance** | CP | $0.61 \pm 0.04$ | 0.55 |
| | AS | $0.66 \pm 0.04$ | 0.62 |
| **SMC** | CP | $0.51 \pm 0.04$ | 0.35 |
| | AS | $0.49 \pm 0.04$ | 0.48 |
| **MPGD** | CP | $0.56 \pm 0.04$ | 0.45 |
| | AS | $0.48 \pm 0.04$ | 0.35 |
| **SVDD** | CP | $0.82 \pm 0.03$ | 0.82 |
| | AS | $0.59 \pm 0.04$ | 0.57 |
| **CTRL** | **CP** | $\mathbf{0.94 \pm 0.02}$ | **0.94** |
| | **AS** | $\mathbf{0.93 \pm 0.02}$ | **0.93** |

(b) Evaluation.

(c) Images generated by **CTRL**.

Figure 2: Results for multi-task conditioning. Figure a plots the histogram of samples generated by the pre-trained diffusion model. Table b presents the evaluation statistics. Figure c displays images generated by a **single** model fine-tuned with **CTRL**.

generating samples conditioned on $Y = 1$ or $Y = 3$ is challenging due to the infrequent occurrence of such samples from the pre-trained model.

**Results.** We follow the evaluation procedure outlined in Section 6.1, with results summarized in Table 2b, demonstrating that **CTRL** outperforms all baselines across both tasks by a big margin. Notably, **CTRL** can generate samples rarely produced by the pre-trained model with over $90\%$ accuracy, particularly for the desired class $Y = 3$ (highly aesthetic images with high compressibility). Producing such images is challenging as aesthetically pleasing images typically require more storage and thus have low compressibility. Generated images are displayed in Figure 2c. More visualizations are provided in Appendix E.5.

## 7 CONCLUSION

We introduce a provable RL-based fine-tuning approach for conditioning pre-trained diffusion models on additional controls. Compared to classifier-free guidance, our proposed method uses the offline dataset more efficiently and is able to leverage the conditional independence assumption, thereby greatly simplifying the construction of the offline dataset. Our approach is empirically validated across three settings: image generation conditioned on a new task, image generation conditioned on the composition of two new tasks, and biological sequence design.

**Reproducibility Statement.** We submit the code for our image experiments as supplementary materials. Complete proofs of our theoretical results are provided in Appendix D. Detailed information about our experiments, including dataset descriptions, model architecture, and baseline implementations, can be found in Appendix E.

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

## A  TRAINING DIFFUSION MODELS

In standard diffusion models, given a training dataset $\{x^{\langle j\rangle}\} \sim p_{\text{data}}(\cdot)$, the goal is to construct a transport that maps noise distribution and data distribution $p_{\text{data}} \in \Delta(\mathcal{X})$ ($\mathcal{X} = \mathbb{R}^d$). More specifically, suppose that we have an SDE [7]:

$$dx_t = f(t, x_t; \theta)dt + \sigma(t)dw_t, \tag{8}$$

where $f : [0, T] \times \mathbb{R}^d \to \mathbb{R}^d$ is a drift coefficient, $\sigma : [0, T] \to \mathbb{R}$ is a diffusion coefficient, $w_t$ is $d$-dimensional Brownian motion, and initial state $x_0 \sim p_{\text{ini}}$ where $p_{\text{ini}} \in \Delta(\mathcal{X})$ denotes the initial distribution. By denoting the marginal distribution at time $T$ by $p_T^\theta(x)$, a standard goal in training diffusion models is to learn the parameter $\theta$ so that $p_T^\theta(x) \approx p_{\text{data}}$. This means we can (approximately) sample from $p_{\text{data}}$ by following the SDE (8) from $0$ to $T$.

To train diffusion models, we first introduce a (fixed) *forward* reference SDE, which gradually adds noise to $p_{\text{data}}$:

$$dz_t = \bar{f}(t, z_t)dt + \bar{\sigma}(t)dw_t, \quad z_0 \sim p_{\text{data}}, \tag{9}$$

where $\bar{f} : [0, T] \times \mathbb{R}^d \to \mathbb{R}^d$ is a drift coefficient, $\bar{\sigma} : [0, T] \to \mathbb{R}$ is a diffusion coefficient. An example is the classical denoising diffusion model (Ho et al., 2020), also known as the variance-preserving (VP) process, which sets $\bar{f} = -0.5z_t, \bar{\sigma} = 1$.

Now, we consider the time-reversal SDE (Anderson, 1982), which reverses the direction of SDE while keeping the marginal distribution, as follows:

$$dx_t = \left\{-\bar{f}(T - t, x_t) + \nabla \log q_{T-t}(x_t)\right\} dt + \bar{\sigma}(T - t)dw_t, \quad x_0 \sim \mathcal{N}(0, I_d). \tag{10}$$

Here, $q_t(\cdot)$ denotes the marginal distribution at time $t$ for the distribution induced by the reference SDE, and $\nabla \log q_{T-t}(x_t)$ means a derivative w.r.t. $x_t$, which is often referred to as the *score function*. Furthermore, when the time horizon $T$ is sufficiently large, $z_t$ follows Gaussian noise distribution $\mathcal{N}(0, I_d)$. Hence, if we could learn the score function, by following the SDE (10) starting from Gaussian noise, we can sample from the data distribution.

Then, we aim to learn the score function from the data. By comparing the time-reversal SDE with the original SDE, a natural parameterization is:

$$f(t, x_t; \theta) = -\bar{f}(T - t, x_t) + s(T - t, x_t; \theta), \quad \sigma(t) = \bar{\sigma}(T - t),$$

where $s(T - t, x_t; \theta)$ is the parametrized neural network introduced to approximate the score function $\nabla \log q_{T-t}(x_t)$. Here, we can leverage the analytical form of the conditional distribution $q_{T-t|0}(\cdot|\cdot)$ (which is a Gaussian distribution derived from the reference SDE). This approach enables us to tackle the approximation problem via regression:

$$\hat{\theta} = \underset{\theta}{\operatorname{argmin}} \, \mathbb{E}_{t \in [0,T], z_0 \sim p_{\text{data}}, z_t \sim q_{t|0}(z_0)} \left[ \lambda(t) \left\| s(t, z_t; \theta) - \nabla_{z_t} \log q_{t|0}(z_t|z_0) \right\|^2 \right], \tag{11}$$

where $\lambda : [0, T] \to \mathbb{R}$ is a weighting function.

---

[7]In standard diffuson models, the direction is reversed, i.e., $x_T$ corresponds to the noise distribution.

## B PLANNING ALGORITHM FOR **CTRL**

---

**Algorithm 2** Direct back-propagation for conditioning

---

1: **Input** Batchsize $n$, Learning rate $\eta$, Discretization step $\Delta t$, Exploratory distribution $\Pi \in \Delta(\mathcal{C} \times \mathcal{Y})$.
2: **Itinialize**: $\psi = [\{\theta^{\mathrm{pre}}\}^\top, \mathbf{0}^\top]$
3: **for** $i \leftarrow 1$ to $S$ **do**
4:     We obtain $n$ trajectories
$$\{X_0^{\langle k \rangle}, \cdots, X_T^{\langle k \rangle}\}_{k=1}^n, \{Z_0^{\langle k \rangle}, \cdots, Z_T^{\langle k \rangle}\}_{k=1}^n.$$
  following $(C^{\langle k \rangle}, Y^{\langle k \rangle}) \sim \Pi(\cdot), X_0^{\langle k \rangle} \sim \mathcal{N}(0, I_d), Z_0 = 0$, and
$$X_t^{\langle k \rangle} = X_{t-1}^{\langle k \rangle} + g(t-1, C^{\langle k \rangle}, Y^{\langle k \rangle}, X_{t-1}^{\langle k \rangle}; \psi_i)\Delta t + \sigma(t)(\Delta w_t), \quad \Delta w_t \sim \mathcal{N}(0, (\Delta t)^2),$$
$$Z_t^{\langle k \rangle} = Z_{t-1}^{\langle k \rangle} + \frac{\|g(t-1, C^{\langle k \rangle}, Y^{\langle k \rangle}, X_{t-1}^{\langle k \rangle}; \psi_i) - f^{\mathrm{pre}}(t-1, C^{\langle k \rangle}, X_{t-1}^{\langle k \rangle}; \theta^{(i)})\|^2}{2\sigma^2(t-1)}\Delta t.$$
5:     Update a parameter:
$$\psi_{i+1} = \psi_i + \eta\nabla_\psi \left\{ \frac{1}{n}\sum_{k=1}^n \left[ \gamma\log\hat{p}(Y^{\langle k \rangle}|X_T^{\langle k \rangle}, C^{\langle k \rangle}) - Z_T^{\langle k \rangle} \right] \right\}\Big|_{\psi=\psi_i},$$
6: **end for**
7: **Output**: Parameter $\psi_S$

---

Inspired by (Clark et al., 2023; Prabhudesai et al., 2023), our planning algorithm, listed in Algorithm 2, is based on direct back-propagation. This method is iterative in nature. During each iteration, we: (1) compute the expectation over trajectories ($\mathbb{E}_{x_{0:T} \sim \mathbb{P}^{g(\cdot;\psi)}}$) using discretization techniques such as Euler-Maruyama; (2) directly optimize the KL-regularized objective function with respect to parameters of the augmented model (i.e., $\psi$).

In practice, such computation might be memory-intensive when there are numerous discretization steps and the diffusion models have a large number of parameters. This is because gradients would need to be back-propagated through the diffusion process. To improve computational efficiency, we recommend employing specific techniques, including (a) only fine-tuning LoRA (Hu et al., 2021) modules instead of the full diffusion weights, (b) employing gradient checkpointing (Gruslys et al., 2016; Chen et al., 2016) to conserve memory, and (c) randomly truncating gradient back-propagation to avoid computing through all diffusion steps (Clark et al., 2023; Prabhudesai et al., 2023).

**Remark 4** (PPO). *In Algorithm 1, we employ direct back-propagation (i.e., Algorithm 2) for planning (i.e., solving the RL problem* (5)*), which necessarily demands the differentiability of the classifier. If the classifier is non-differentiable, we suggest using Proximal Policy Optimization (PPO) for planning, such as Schulman et al. (2017); Black et al. (2023); Fan et al. (2023). Other parts remain unchanged.*

## C INFERENCE TECHNIQUE IN CLASSFIER-FREE GUIDANCE

Although the fine-tuning process sets the guidance level for the additional conditioning (i.e., $y$) at a specific $\gamma$, classifier-free guidance makes it possible to adjust the guidance strength freely during inference. Recall that the augmented model is constructed as: $g(t, c, y, x; \psi)$ where $\psi = [\theta^\top, \phi^\top]^\top$. Suppose we have obtained a drift term $\hat{g}$, parametrized by $\hat{\psi} = [\hat{\theta}^\top, \hat{\phi}^\top]^\top$ from running Algorithm 1. In inference, we may alter the guidance levels by using the following drift term in the SDE (3)

$$g_{\gamma_1,\gamma_2}(t, c, y, x_t)$$
$$= \underbrace{g(t, \emptyset, \emptyset, x_t; \hat{\psi}) + \gamma_1(g(t, c, \emptyset, x_t; \hat{\psi}) - g(t, \emptyset, \emptyset, x_t; \hat{\psi}))}_{\text{Term 1: pre-trained diffusion model conditioned on } \mathcal{C}} + \underbrace{\gamma_2(g(t, c, y, x_t; \hat{\psi}) - g(t, c, \emptyset, x_t; \hat{\psi}))}_{\text{Term 2: additional conditioning on } \mathcal{Y}}$$

where $\emptyset$ indicates the unconditional on $\mathcal{Y}$ or on $\mathcal{C}$. In the above, both $\gamma_1$ and $\gamma_2$ do not necessarily need to equal $\gamma$. They can be adjusted respectively to reflect guidance strength levels for two conditions.

# D  PROOFS

## D.1  IMPORTANT LEMMAS

We first introduce several important lemmas to prove our main statement.

First, recall that $\mathbb{P}^g(\cdot|c, y)$ is the induced distribution by the SDE:

$$dx_t = g(t, c, y, x_t)dt + \sigma(t)dw_t, \quad x_0 = x_{\mathrm{ini}}$$

over $\mathcal{K}$ conditioning on $c$ and $y$. Similarly, denote $\mathbb{P}^{\mathrm{pre}}(\cdot|c)$ by the induced distribution by the SDE:

$$dx_t = f^{\mathrm{pre}}(t, c, x_t)dt + \sigma(t)dw_t, \quad x_0 = x_{\mathrm{ini}}$$

over $\mathcal{K}$ conditioning on $c$.

**Lemma 3** (KL-constrained reward). *The objective function in* (5) *is equivalent to*

$$\mathrm{obj} = \mathbb{E}_{(c,y)\sim\Pi,\mathbb{P}^g(\cdot|c,y)}[\gamma\log p^\diamond(y|x_T, c) - \mathrm{KL}(\mathbb{P}^g(\cdot|c, y)\|\mathbb{P}^{\mathrm{pre}}(\cdot|c)]. \tag{12}$$

*Proof.* We calculate the KL divergence of $\mathbb{P}^g$ and $\mathbb{P}^{\mathrm{pre}}$ as below

$$\mathrm{KL}(\mathbb{P}^g(\cdot|c,y)\|\mathbb{P}^{\mathrm{pre}}(\cdot|c) = \mathbb{E}_{x_{0:T}\sim\mathbb{P}^g(\cdot|c,y)}\left[\int_0^T \frac{1}{2}\frac{\|g(t,c,y,x_t) - f^{\mathrm{pre}}(t,c,x_t)\|^2}{\sigma^2(t)}dt\right]. \tag{13}$$

This is because

$$\mathrm{KL}(\mathbb{P}^g(\cdot|c,y)\|\mathbb{P}^{\mathrm{pre}}(\cdot|c))$$

$$= \mathbb{E}_{\mathbb{P}^g(\cdot|c,y)}\left[\frac{d\mathbb{P}^g(\cdot|c,y)}{d\mathbb{P}^{\mathrm{pre}}(\cdot|c)}\right]$$

$$= \mathbb{E}_{\mathbb{P}^g(\cdot|c,y)}\left[\int_0^T \frac{1}{2}\frac{\|g(t,c,y,x_t) - f^{\mathrm{pre}}(t,c,x_t)\|^2}{\sigma^2(t)}dt + \int_0^T \{g(t,c,y,x_t) - f^{\mathrm{pre}}(t,c,x_t)\}dw_t\right]$$

$$\text{(Girsanov theorem)}$$

$$= \mathbb{E}_{\mathbb{P}^g(\cdot|c,y)}\left[\int_0^T \frac{1}{2}\frac{\|g(t,c,y,x_t) - f^{\mathrm{pre}}(t,c,x_t)\|^2}{\sigma^2(t)}dt\right]. \quad \text{(Martingale property of Itô integral)}$$

Therefore, the objective function in (5) is equivalent to

$$\mathrm{obj} = \mathbb{E}_{(c,y)\sim\Pi,\mathbb{P}^g(\cdot|c,y)}[\gamma\log p^\diamond(y|x_T, c) - \mathrm{KL}(\mathbb{P}^g\|\mathbb{P}^{\mathrm{pre}})]. \tag{14}$$

$\square$

**Optimal value function.**  For the RL problem (5), it is beneficial to introduce the optimal optimal value function $v_t^\star(x|c, y)$ at any time $t \in [0, T]$, given $x_t = x$, conditioned on parameters $c$ and $y$ defined as:

$$v_t^\star(x|c, y) = \max_g \mathbb{E}\left[\gamma\log p^\diamond(y|x_T, c) - \frac{1}{2}\int_t^T \frac{\|f^{\mathrm{pre}}(s, c, x_s) - g(s, c, y, x_s)\|^2}{\sigma^2(s)}ds \,\Big|\, x_t = x, c, y\right]. \tag{15}$$

Specifically, we note that $v_T(x|c, y) = \gamma\log p^\diamond(y|x, c)$ represents the terminal reward function (i.e., a loglikelihood in our MDP), while $v_0$ represents the original objective function (5) that integrates the entire trajectory's KL divergence along with the terminal reward.

Below we derive the optimal value function in analytical form.

**Lemma 4** (Feynman–Kac Formulation). *At any time $t \in [0, T]$, given $x_t = x$, and conditioned on $c$ and $y$, we have the optimal value function $v_t^*(x|c, y)$ (induced by the optimal drift term $g^*$) as follows*

$$\exp\left(v_t^\star(x|c, y)\right) = \mathbb{E}_{\mathbb{P}^{\mathrm{pre}}(\cdot|c)}\left[(p^\diamond(y|x_T, c))^\gamma | x_t = x, c\right].$$

*Proof.* From the Hamilton–Jacobi–Bellman (HJB) equation, we have

$$\max_u \left\{ \frac{\sigma^2(t)}{2} \sum_i \frac{d^2 v_t^\star(x|c, y)}{dx^{[i]} dx^{[i]}} + g \cdot \nabla v_t^\star(x|c, y) + \frac{dv_t^\star(x|c, y)}{dt} - \frac{\|g - f^{\mathrm{pre}}\|_2^2}{2\sigma^2(t)} \right\} = 0. \quad (16)$$

where $x^{[i]}$ is a $i$-th element in $x$. Hence, by simple algebra, we can prove that the optimal drift term satisfies

$$g^\star(t, c, y, x) = f^{\mathrm{pre}}(t, c, x) + \sigma^2(t)\nabla v_t^\star(x|c, y).$$

By plugging the above into the HJB equation (16), we get

$$\frac{\sigma^2(t)}{2} \sum_i \frac{d^2 v_t^\star(x|c, y)}{dx^{[i]} dx^{[i]}} + f^{\mathrm{pre}} \cdot \nabla v_t^\star(x|c, y) + \frac{dv_t^\star(x|c, y)}{dt} + \frac{\sigma^2(t)\|\nabla v_t^\star(x|c, y)\|_2^2}{2} = 0, \quad (17)$$

which characterizes the optimal value function. Now, using (17), we can show

$$\frac{\sigma^2(t)}{2} \sum_i \frac{d^2 \exp(v_t^\star(x|c, y))}{dx^{[i]} dx^{[i]}} + f^{\mathrm{pre}} \cdot \nabla \exp(v_t^\star(x|c, y)) + \frac{d \exp(v_t^\star(x|c, y))}{dt}$$

$$= \exp\left(v_t^\star(x|c, y)\right) \times \left\{ \frac{\sigma^2(t)}{2} \sum_i \frac{d^2 v_t^\star(x|c, y)}{dx^{[i]} dx^{[i]}} + f^{\mathrm{pre}} \cdot \nabla v_t^\star(x|c, y) + \frac{dv_t^\star(x|c, y)}{dt} + \frac{\sigma^2(t)\|\nabla v_t^\star(x|c, y)\|_2^2}{2} \right\}$$

$$= 0.$$

Therefore, to summarize, we have

$$\frac{\sigma^2(t)}{2} \sum_i \frac{d^2 \exp(v_t^\star(x|c, y))}{dx^{[i]} dx^{[i]}} + f^{\mathrm{pre}} \cdot \nabla \exp(v_t^\star(x|c, y)) + \frac{d \exp(v_t^\star(x|c, y))}{dt} = 0, \quad (18)$$

$$v_T^\star(x|c, y) = \gamma \log p^\diamond(y|x, c). \quad (19)$$

Finally, by invoking the Feynman-Kac formula (Shreve et al., 2004), we obtain the conclusion:

$$\exp\left(v_t^\star(x|c, y)\right) = \mathbb{E}_{\mathbb{P}^{\mathrm{pre}}(\cdot|x_t, c)}\left[(p^\diamond(y|x_T, c))^\gamma | x_t = x, c\right].$$

$\square$

## D.2 Proof of Theorem 1

Firstly, we aim to show that the optimal conditional distribution over $\mathcal{K}$ on $c$ and $y$ (i.e., $\mathbb{P}^{g^*}(\tau|c, y)$) is equivalent to

$$\frac{\mathbb{P}^{\mathrm{pre}}(\tau|c)(p^\diamond(y|x_T, c))^\gamma}{C(c, y)}, \quad C(c, y) := \exp(v_0^\star(x_0|c, y)).$$

To do that, we need to check that the above is a valid distribution first. This is indeed valid because the above is decomposed into

$$\underbrace{\frac{(p^\diamond(y|x_T, c))^\gamma \cdot \mathbb{P}^{\mathrm{pre}}(x_T|c)}{C(c, y)}}_{(\alpha 1)} \times \underbrace{\mathbb{P}^{\mathrm{pre}}(\tau|c, x_T)}_{(\alpha 2)}, \quad (20)$$

and both $(\alpha 1), (\alpha 2)$ are valid distributions. Especially, for the term $(\alpha 1)$, we observe

$$C(c, y) = \int (p^\diamond(y|x_T, c))^\gamma d\mathbb{P}^{\mathrm{pre}}(x_T|c)) = \mathbb{E}_{\mathbb{P}^{\mathrm{pre}}(\cdot|c)}[(p^\diamond(y|x_T, c))^\gamma] = \exp(v_0^\star(x_0|c, y)).$$

(cf. Lemma 4)

Now, after checking (20) is a valid distribution, we calculate the KL divergence:

$$\mathrm{KL}\left(\mathbb{P}^{g^\star}(\tau|c,y)\middle\|\frac{\mathbb{P}^{\mathrm{pre}}(\tau|c)(p^\diamond(y|x_T,c))^\gamma}{C(c,y)}\right)$$

$$= \mathrm{KL}(\mathbb{P}^{g^\star}(\tau|c,y)\|\mathbb{P}^{\mathrm{pre}}(\tau|c)) - \mathbb{E}_{\mathbb{P}^{g^\star}(\cdot|c,y)}\left[\gamma \log p^\diamond(y|x_T,c) - \log C(c,y)\right]$$

$$= \mathbb{E}_{\mathbb{P}^{g^\star}(\cdot|c,y)}\left[\left\{\int_0^T \frac{1}{2}\frac{\|g^\star(t,c,y,x_t) - f^{\mathrm{pre}}(t,c,x_t)\|^2}{\sigma^2(t)}\right\}dt - \gamma \log p^\diamond(y|x_T,c) + \log C(c,y)\right]$$
(cf. KL divergence (13))

$$= -v_0^\star(x_0|c,y) + \log C(c,y).$$  (Definition of optimal value function)

Therefore,

$$\mathrm{KL}\left(\mathbb{P}^{g^\star}(\tau|c,y)\middle\|\frac{\mathbb{P}^{\mathrm{pre}}(\tau|c)(p^\diamond(y|x_T,c))^\gamma}{C(c,y)}\right) = -v_0^\star(x_0|c,y) + \log C(c,y) = 0.$$

Hence,

$$\mathbb{P}^{g^\star}(\tau|c,y) = \frac{\mathbb{P}^{\mathrm{pre}}(\tau|c)(p^\diamond(y|x_T,c))^\gamma}{C(c,y)}.$$

**Marginal distribution at $t$.** Finally, consider the marginal distribution at $t$. By marginalizing before $t$, we get

$$\mathbb{P}^{\mathrm{pre}}(\tau_{[t,T]}|c) \times (p^\diamond(y|x_T,c))^\gamma/C(c,y).$$

Next, by marginalizing after $t$,

$$\mathbb{P}_t^{\mathrm{pre}}(x|c)/C(c,y) \times \mathbb{E}_{\mathbb{P}^{\mathrm{pre}}(\cdot|c)}[(p^\diamond(y|x_T,c))^\gamma|x_t = x, c].$$

Using Feynman–Kac formulation in Lemma 4, this is equivalent to

$$\mathbb{P}_t^{\mathrm{pre}}(x|c)\exp(v_t^\star(x|c,y))/C(c,y).$$

**Marginal distribution at $T$.** We marginalize before $T$. We have the following

$$\mathbb{P}_T^{\mathrm{pre}}(x|c)(p^\diamond(y|x_T,c))^\gamma/C(c,y).$$

### D.3 PROOF OF LEMMA 2

Recall $g^\star(t,c,y,x) = f^{\mathrm{pre}}(t,c,x) + \sigma^2(t) \times \nabla_x v_t^\star(x|c,y)$ from the proof of Lemma 4, we have

$$g^\star(t,c,y,x) = f^{\mathrm{pre}}(t,c,x) + \sigma^2(t) \times \nabla_x \log \mathbb{E}_{\mathbb{P}^{\mathrm{pre}}(\cdot|c)}[(p^\diamond(y|x_T,c))^\gamma|x_t = x, c].$$

## E DETAILS OF EXPERIMENTS

### E.1 IMPLEMENTATION OF CLASSIFIER-FREE BASELINE

The effectiveness of classifier-free guidance often relies on a sufficiently large offline dataset $(c,x,y)$. However, in our experiments (Section 6.1 and Section 6.2), we only have access to offline datasets $(x,y)$ and $(x,y_1,y_2)$ respectively. Thus, to implement a classifier-free guidance baseline in this context, we leverage the pre-trained diffusion model for data augmentation. The procedure for Section 6.1 is outlined below. The procedure for Section 6.2 is similar.

**Data augmentation.** Consider the scenario where $Y \perp C|X$ and we have access to $p^{\mathrm{pre}}$ and offline dataset $\{(x,y)\}$. First, we use the offline data $\{(x,y)\}$ to train a classifier $\hat{p} : \mathcal{X} \to \Delta(\mathcal{Y})$. We then use $\hat{p}$ to generate triplets $(x,c,y) \sim p^{\mathrm{pre}}(x|c)\hat{p}(y|x)$ for given $c$. In practice, for text-to-image diffusion models, $c$ can be uniformly sampled from a set of prompts, such as animals. However, this process becomes computationally demanding when applied to large pre-trained models like Stable Diffusion(Rombach et al., 2022).

**Potential limitations.** Given the data augmentation strategy, several limitations arise for classifier-free guidance. A primary concern is the accuracy of the trained classifier $\hat{p}(y|x)$. If the classifier is not sufficiently accurate, the generated $y$ values may be unreliable, compromising the quality of the augmented triplets $(x, c, y)$. Additionally, selecting the condition $c$ presents challenges. While models like Stable Diffusion (Rombach et al., 2022) are pre-trained on vast and diverse datasets with a wide range of prompts, we are constrained to a smaller, more limited set of prompts for $c$ in this context. This lack of diversity reduces the representativeness of the augmented data and may lead to mode collapse during fine-tuning—a common issue observed in fine-tuning of diffusion models (Uehara et al., 2024).

### E.2 Implementation of Reconstruction Guidance Baseline

As reviewed in Section 3.2.2, reconstruction guidance baseline employs the following approximation

$$p(y|x_t, c) = \int p^\diamond(y|x_T, c)p(x_T|x_t, c)dx_T \approx p(y|\hat{x}_T(x_t, c), c),$$

where $\hat{x}_T(x_t, c)$ is the (expected) denoised sample given $x_t, c$, i.e., $\hat{x}_T(x_t, c) = \mathbb{E}[x_T|x_t, c]$. We note that such approximation is often readily available from diffusion noise schedulers, such as DDIM scheduler (Song et al., 2020).

Given such an approximation, we only need to learn $p^\diamond(y|x_T, c)$ from offline data. Accordingly, we can leverage the trained classifiers from Algorithm 1 (see Section 4.2, **Step 2**).

As an inference-time technique, the choice of guidance strength is often subtle. We present ablation studies on guidance strength in Appendix E.4. For reporting classification metrics, as shown in Table 1b and Table 2b, we consistently select the optimal configuration for each baseline.

### E.3 Images

In this subsection, we provide details of experiments in Section 6. We first explain the training details and list hyperparameters in Table 2.

We use 4 A100 GPUs for all the image tasks. We use the AdamW optimizer (Loshchilov and Hutter, 2019) with $\beta_1 = 0.9, \beta_2 = 0.999$ and weight decay of $0.1$. To ensure consistency with previous research, in fine-tuning, we also employ training prompts that are uniformly sampled from 50 common animals (Black et al., 2023; Prabhudesai et al., 2023).

Table 2: Training hyperparameters.

| Hyperparameter | compressibility (Section 6.1) | multi-task (Section 6.2) |
|---|---|---|
| Classifier-free guidance weight on prompts (i.e., $c$) | 7.5 | 7.5 |
| $\gamma$ (i.e., strength of the additional guidance on $y$) | 10 | 10 |
| DDIM steps | 50 | 50 |
| Truncated back-propagation step | $K \sim \text{Uniform}(0, 50)$ | $K \sim \text{Uniform}(0, 50)$ |
| Learning rate for LoRA modules | $1e^{-3}$ | $3e^{-4}$ |
| Learning rate for the linear embeddings | $1e^{-2}$ | $1e^{-2}$ |
| Batch size (per gradient update) | 256 | 512 |
| Number of gradient updates per epoch | 2 | 2 |
| Epochs | 15 | 60 |

**Construction of the augmented score model.** An important engineering aspect is how to craft the augmented score model architecture. For most of the diffusion models, the most natural and direct technique of adding another conditioning control is (1) augmenting the score prediction networks by incorporating additional linear embeddings, while using the existing neural network architecture and weights for all other parts. In our setting, we introduce a linear embedding layer that maps $|\mathcal{Y}| + 1$ class labels to embeddings in $\mathbb{R}^d$, where $d$ is the same dimension as intermediate diffusion states. Among all embeddings, the first $|\mathcal{Y}|$ embeddings correspond to $|\mathcal{Y}|$ conditions of our interest, whereas the last one represents the unconditional category (i.e., NULL conditioning) (2) for any $y \in \mathcal{Y}$, the corresponding embedding is added to the predicted score in the forward pass. During fine-tuning, the embeddings are initialized as zeros. We only fine-tune the first $|\mathcal{Y}|$ embeddings, and freeze the last one at zero as it is the unconditional label.

We note that, while it is possible to add additional conditioning by reconstructing the score networks like ControlNet (Zhang et al., 2023), in practice it is often desired to make minimal changes to the architecture of large diffusion models, e.g., Stable Diffusion (Rombach et al., 2022) to avoid the burdensome re-training. It is especially important to leverage pre-trained diffusion models in our setting where the offline dataset is limited, therefore a total retraining of model parameters can be struggling.

**Sampling.** We use the DDIM sampler with $50$ diffusion steps (Song et al., 2020). Since we need to back-propagate the gradient of rewards through both the sampling process producing the latent representation and the VAE decoder used to obtain the image, memory becomes a bottleneck. We employ two designs to alleviate memory usage following Clark et al. (2023); Prabhudesai et al. (2023): (1) Fine-tuning low-rank adapter (LoRA) modules (Hu et al., 2021) instead of tuning the original diffusion weights, and (2) Gradient checkpointing for computing partial derivatives on demand (Gruslys et al., 2016; Chen et al., 2016). The two designs make it possible to back-propagate gradients through all 50 diffusing steps in terms of hardware.

Table 3: Architecture of compressibility classifier.

| # | Input Dimension | Output Dimension | Layer |
|---|---|---|---|
| 1 | $C \times H \times W$ | $64 \times H \times W$ | ResidualBlock (Conv2d(3, 64, 3x3), BN, ReLU) |
| 2 | $64 \times H \times W$ | $128 \times \frac{H}{2} \times \frac{W}{2}$ | ResidualBlock (Conv2d(64, 128, 3x3), BN, ReLU) |
| 3 | $128 \times \frac{H}{2} \times \frac{W}{2}$ | $256 \times \frac{H}{4} \times \frac{W}{4}$ | ResidualBlock (Conv2d(128, 256, 3x3), BN, ReLU) |
| 4 | $256 \times \frac{H}{4} \times \frac{W}{4}$ | $256 \times 1 \times 1$ | AdaptiveAvgPool2d (1, 1) |
| 5 | $256 \times 1 \times 1$ | $256$ | Flatten |
| 6 | $256$ | num_classes | Linear |

Table 4: Architecture of aesthetic score classifier.

| # | Layer | Input Dimension | Output Dimension |
|---|---|---|---|
| 1 | Linear | 768 | 1024 |
| 2 | Dropout | - | - |
| 3 | Linear | 1024 | 128 |
| 4 | Dropout | - | - |
| 5 | Linear | 128 | 64 |
| 6 | Dropout | - | - |
| 7 | Linear | 64 | 16 |
| 8 | Linear | 16 | num_classes |

**Classifiers.** In our experiments, we leverage conditional independence for both compressibility and aesthetic scores tasks. Therefore, we only demand data samples $\{x_i, y_i\}$ in order to approximate conditional classifier $p(y|x, c)$. Specifically,

- compressibility: the classifier is implemented as a 3-layer convolutional neural network (CNN) with residual connections and batch normalizations on top of the raw image space. The offline dataset is constructed by labeling a subset of 10k images of the AVA dataset (Murray et al., 2012), employing JPEG compression. We train the network using Adam optimizer for 100 epochs. Detailed architecture of the oracle can be found in Table 3.

- aesthetic scores: the classifier is implemented as an MLP on top of CLIP embeddings (Radford et al., 2021). To train the classifier, we use the full AVA dataset (Murray et al., 2012) which includes more than 250k human evaluations. The specific neural network instruction is listed in Table 4.

Note that in training both classifiers, we split the dataset with $80\%$ for training and $20\%$ for validation. After training, we use the validation set to perform temperature scaling calibration Guo et al. (2017).

## E.4 ADDITIONAL RESULTS OF RECONSTRUCTION GUIDANCE BASELINE

In this subsection, we provide more results of **Reconstruction Guidance** for conditioning images on compressibility (see Section 6.1).

In Figure 3, we plot the confusion matrix for samples generated by **Reconstruction Guidance**. For each condition, 128 samples are generated and are evaluated. We find that this method struggles to generate samples accurately when conditioned on intermediate labels.

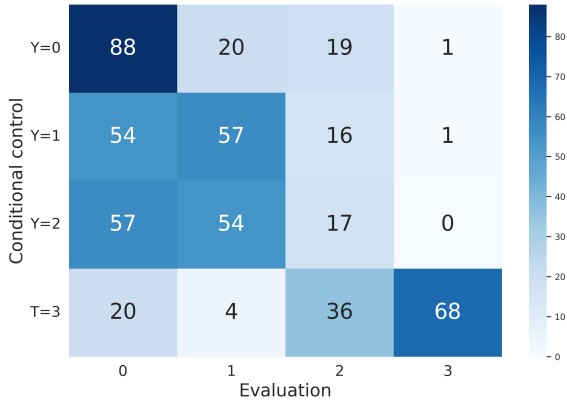

Figure 3: Confusion matrix for **Reconstruction Guidance**.

Table 5: Results of **Reconstruction Guidance** conditioned on compressibility. Essentially, guidance level$= 0$ indicates that the generations are unconditional on $Y$.

| Conditional control ($Y$) | Guidance level ($\gamma$) | Accuracy ↑ | Mean score |
|---|---|---|---|
| 0 | 0 | 0.43 | −110 |
| 0 | 5 | 0.64 | −157.4 |
| 0 | 7.5 | 0.62 | −148.2 |
| 0 | 10 | 0.52 | −156.5 |
| 0 | 20 | 0.66 | −152.5 |
| 0 | 50 | 0.69 | −151.6 |
| 1 | 0 | 0.45 | −110 |
| 1 | 5 | 0.14 | −152.0 |
| 1 | 7.5 | 0.12 | −189.7 |
| 1 | 10 | 0.08 | −163.1 |
| 1 | 20 | 0.06 | −169.5 |
| 1 | 50 | 0 | −194.0 |
| 2 | 0 | 0.13 | −110 |
| 2 | 5 | 0.02 | −104.9 |
| 2 | 7.5 | 0.10 | −122.1 |
| 2 | 10 | 0.12 | −111.3 |
| 2 | 20 | 0.08 | −157.6 |
| 2 | 50 | 0.08 | −173.5 |
| 3 | 0 | 0 | −110 |
| 3 | 5 | 0.46 | −71.7 |
| 3 | 7.5 | 0.46 | −67.6 |
| 3 | 10 | 0.53 | −65.6 |
| 3 | 20 | 0.26 | −112.4 |
| 3 | 50 | 0.32 | −121.5 |

For completeness, below we provide ablation studies on its hyper-parameters. In Table 5, we present the classification statistics for generations across different conditions ($y$) and guidance levels ($\gamma$). Recall that the four conditions are defined as follows: $Y = 0 :$ **CP** $< -110.0$, $Y = 1 :$ $-110.0 \leq$ **CP** $< -85.0$, $Y = 2 :$ $-85.0 \leq$ **CP** $< -60.0$, $Y = 3 :$ **CP** $\geq -60.0$.

Our analysis reveals several key insights:

1. For $Y = 0$ and $Y = 3$, the accuracy of the generations improves as the guidance signal strength increases. This indicates a clear positive correlation between the guidance level and the accuracy of generation.

2. Conversely, for intermediate $Y = 1$ and $Y = 2$, guidance signals decrease generation accuracy compared to the pre-trained model, suggesting difficulty in maintaining accuracy within these specific compressibility intervals. The challenge in generating samples with medium compressibility scores lies in hand-picking the guidance strength. For instance, generating samples conditioned on $Y = 2$ requires compressibility scores between $-85$ and $-60$, making it difficult to apply optimal guidance without overshooting or undershooting the target values.

3. Regarding the mean scores, distinct patterns are observed across different conditions and guidance levels:
   - For $Y = 0$, mean scores become more negative with increasing guidance levels.
   - For $Y = 1$, mean scores consistently drop with increasing guidance levels.
   - For $Y = 2$, mean scores initially improve slightly with increasing guidance levels but show a marked decline at $\gamma = 20$ and $\gamma = 50$, indicating a challenge in achieving the desired compressibility range.
   - For $Y = 3$, mean scores improve significantly with increased guidance, showing the best results at $\gamma = 10$, but then become more negative at higher guidance levels.

In summary, these observations suggest that while guidance can be beneficial for improving accuracy in extreme compressibility levels ($Y = 0$ and $Y = 3$), this method struggles with intermediate conditions ($Y = 1$ and $Y = 2$) due to the narrow range of acceptable scores and the non-linear effects of guidance strength on generation quality.

For each conditional control, samples are generated by choosing the best $\gamma$ according to Table 5. We report the evaluation statistics in Table 1b, and provide the confusion matrix in Figure 3

### E.5 ADDITIONAL VISUALIZATIONS

We provide more generated samples to illustrate the performances of **CTRL** in Figure 4 and Figure 5.

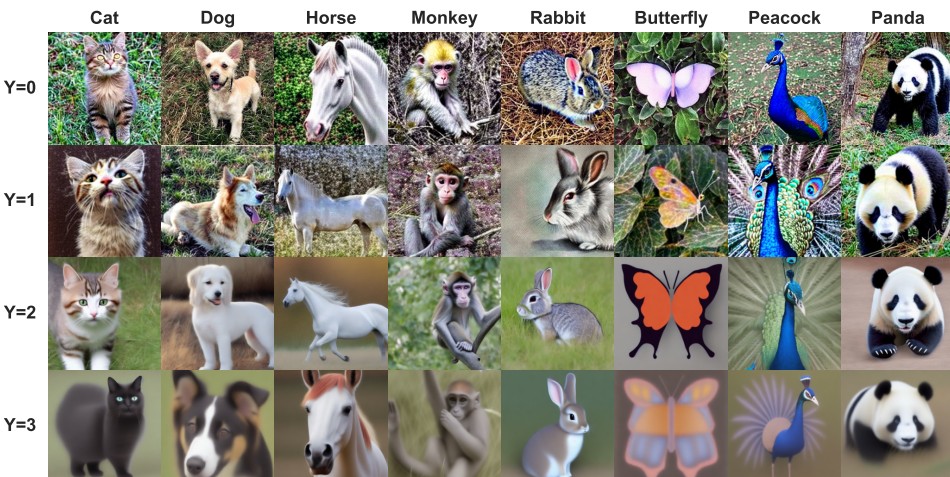

Figure 4: More images generated by **CTRL** in the compressibility task.

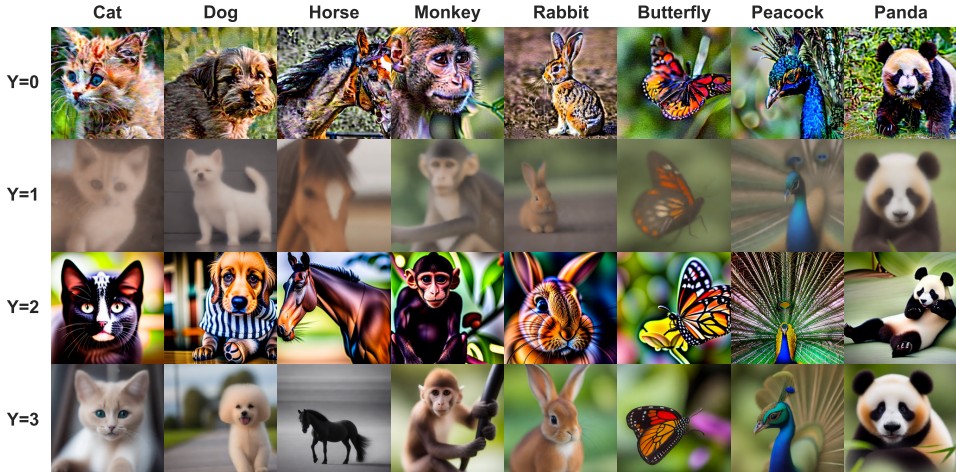

Figure 5: More images generated by **CTRL** in the multi-task conditional generation.

## F  LIMITATIONS AND MITIGATION STRATEGIES

### F.1  COMPUTATIONAL COST

Fine-tuning large pre-trained diffusion models is computationally intensive, especially in multi-task settings. For instance, even in simpler cases, such as fine-tuning a model with a single reward function, the process can be slow—DDPO (Black et al., 2023) requires approximately 60 A100 GPU hours for fine-tuning Stable Diffusion (Rombach et al., 2022). In contrast, our single-task experiment Section 6.1, which conditions on four compressibility labels (a more complex task than optimizing a single reward), is more computationally efficient, requiring only around 20 A100 GPU hours. This demonstrates that our method is computationally efficient even when handling more challenging conditioning tasks.

Multi-task experiments Section 6.2 are considerably more demanding, requiring around 200 A100 GPU hours. This is due to the added complexity of aligning the model with multiple control signals, often necessitating larger batch sizes and careful balancing between tasks. As a result, multi-task fine-tuning requires not only more GPU time but also careful balancing to prevent overfitting to specific tasks.

Below, we discuss promising ways to help reduce the computational burden without sacrificing performance.

**Mitigation strategies.**  To further reduce computational costs, an effective strategy is to truncate backpropagation to a small fixed number of steps, such as 3 or 5. As noted by (Clark et al., 2023), truncating the gradient flow in direct backpropagation to fewer than 10 steps not only significantly reduces computational overhead but also improves optimization stability by mitigating gradient explosion. Interestingly, performance begins to degrade when the number of steps exceeds 10, suggesting that shorter truncation steps (even as few as 1) can be more computationally efficient while maintaining or even improving model performance.

Additionally, mixed precision training can be employed to further accelerate training.

### F.2  MEMORY COMPLEXITY

As we have clarified in Section 4, many types of off-the-shelf RL algorithms can be used for planning. We recommend using direct back-propagation (Clark et al., 2023) or PPO (Black et al., 2023).

For direct backpropagation, updating a single gradient requires $O(L)$ memory, where $L$ is the number of discretizations. To reduce memory usage, our experiments employed techniques such as (a) fine-tuning only LoRA (Hu et al., 2021) modules instead of the full diffusion model, (b) applying

gradient checkpointing (Gruslys et al., 2016; Chen et al., 2016), and (c) randomly truncating gradient backpropagation. A detailed discussion of these techniques is provided in Appendix B.

If memory constraints persist, we recommend using PPO for planning, as it requires only $O(1)$ memory per gradient update. Employing mixed precision training can also reduce memory usage.

### F.3 CHOICE OF GUIDANCE STRENGTH $\gamma$

First, note that $1/\gamma$ can be interpreted as the KL weight parameter in standard diffusion model fine-tuning works (Fan et al., 2023; Uehara et al., 2024), where selecting an optimal KL weight remains an open problem. As observed in these works, fine-tuning without entropy regularization often leads to over-optimization. Therefore, introducing a KL weight is beneficial as long as it is neither too small nor too large.

In this work, a larger $\gamma$ strengthens the guidance signal of the additional control (see Section 4) but can cause the fine-tuned model to deviate more from the pre-trained model, which is also undesired. For both image experiments, we set $\gamma = 10$ (see Table 2), which provides a good balance. In practice, we find that values between $5$ and $20$ are generally effective. Additionally, even if a smaller $\gamma$ is used during fine-tuning, we note that it is possible to freely adjust (strengthen or weaken) the guidance strength during inference. Details can be found in Appendix C.

