# OpenReview forum: "Adding Conditional Control to Diffusion Models with Reinforcement Learning"
_ICLR.cc/2025/Conference — ICLR 2025 Poster_

### Official Review · Reviewer_Po57 · 2024-10-26

**Soundness:** 3
**Presentation:** 3
**Contribution:** 3
**Rating:** 6
**Confidence:** 2

**Summary:**

This paper presents a conditional image generation method based on the diffusion model conditions with classifier guidance. Different from other methods of conditional generation of diffusion model in classifier guidance, this paper uses reinforcement learning to optimize conditional image generation of the diffusion model to maximize the log-likelihood probability of conditional generation as the reward function. It eliminates the need to train the classifier according to the intermediate diffusion frame and reduces the difficulty of training the classifier.

Note that as a researcher in reinforcement learning, I pay more attention to the problems related to RL in this paper. And I'm not familiar with image conditional generation.

**Strengths:**

1. This paper builds a good theoretical framework for RL-based conditional control to diffusion model.
2.  As far as I can see, the idea of using reinforcement learning to fine-tune the generation conditions of diffusion models is interesting.
3. The source code is attached in supplementary material, and I believe it is easy to reproduce.

**Weaknesses:**

1. The experimental part of the paper is very limited, only comparing the two methods in 2022. However, the field of conditional diffusion generation is changing rapidly, and this paper lacks a comparison with the latest methods. This paper does not perform an ablation analysis of the proposed method.
2. The author's specific methods of using RL are not very clear. Only the reward function is introduced, but how to formalize the problem into MDP, the state, the action, the transfer function, and so on are not introduced.

**Questions:**

1. Obviously, CTRL is a splitter-based method, but why is the experimental verification presented as the main comparison with the method without a classifier? Is this unfair? Please explain the considerations behind this.
2. Are there comparisons with more advanced conditional generation models?
3. The authors claim in Appendix G.2 that the proposed method can be adapted to any off-the-shelf RL algorithm, which I take with a grain of salt. There are many kinds of RL algorithms, and the two examples given by the author are both model-free on-policy methods. Did the author take into account offline RL algorithms, model-based RL algorithms, etc.? Is there a more detailed explanation, analysis, or experiment on this?
4. If possible, it is recommended to include a citation to the paper proposing the PPO method [1].

[1] Schulman J, Wolski F, Dhariwal P, et al. Proximal policy optimization algorithms[J]. arXiv preprint arXiv:1707.06347, 2017.

---

> ### Author Response · Authors · 2024-11-21
> **Response**
>
> We sincerely appreciate the reviewer’s acknowledgment of the novelty, theoretical contributions, and reproducibility of our work. Your detailed and thoughtful feedback is greatly valued. In this response, we have addressed your questions and concerns by:
>
> 1. providing additional comparisons with more advanced conditional guidance methods and clarifying the ablation analysis we have conducted
> 2. adding Appendix B, which provides a detailed overview of the formulation for fine-tuning diffusion models using reinforcement learning
> 3. providing clarifications on the comparison between our algorithm and methods without a classifier
> 4. updating the wording of “any off-the-shelf RL algorithms”
> 5. adding a citation to the original PPO paper
>
> **Please note that we have updated a new version of the manuscript with the changes highlighted in red.** We are happy to answer more if further clarification is needed.
>
> ---
>
> **Q. This paper lacks a comparison with the latest conditional methods.**
>
> Thank you for pointing this out! We have included additional baselines for the 4-class compressibility task (Section 6.1) and the multi-task conditional generation (Section 6.2). Below, we briefly introduce the added baseline methods, all of which have used the same trained classifier adopted in CTRL for guidance:
>
> 1. **Sequential Monte Carlo (SMC)-based method**: Recent works [1,2] leverage resampling techniques in SMC to approximate distributions in diffusion models across a batch of samples (i.e., particle filtering). This method is training-free.
>
> 2. **SVDD** [3]: A decoding-based method that iteratively selects preferable samples during each diffusion step based on reward signals. For our conditioning tasks, the compressibility classifier and aesthetic score classifier provide such rewards. This method is also training-free and operates at inference time.
>
> 3. **MPGD** [4]: This method refines the predicted clean data $x_0$​ during inference based on a manifold hypothesis. While such refinement incurs additional computational costs during inference, it does not fine-tune diffusion model weights, remaining a training-free approach.
>
> We report the results below, showing that CTRL achieves good performance, outperforming all baselines in both tasks.
>  While SVDD and MPGD surpass SMC, they still fall short of CTRL. These results highlight CTRL’s superiority in achieving accurate and controllable generation, even compared to the most advanced conditioning methods.
>
> * 4-class compressibility task (Section 6.1)
>
> |                   | Accuracy ↑ | Macro F1 Score ↑ |
> |--------------------------|------------|-------------------|
> | Classifier-Free | 0.33       | 0.28              |
> | Reconstruction Guidance  | 0.45       | 0.45              |
> | SMC                      | 0.27       | 0.22              |
> | MPGD                     | 0.70       | 0.71              |
> | SVDD                     | 0.78       | 0.78              |
> | CTRL                     | 1.0        | 1.0               |
>
> * multi-task conditional generation (Section 6.2)
>
> |                   | Reward | Accuracy ↑ | Macro F1 Score ↑ |
> |--------------------------|---------|------------|-------------------|
> | Classifier-Free | CP      | 0.52       | 0.39              |
> |                          | AS      | 0.59       | 0.55              |
> | Reconstruction Guidance  | CP      | 0.61       | 0.55              |
> |                          | AS      | 0.66       | 0.62              |
> | SMC                      | CP      | 0.51       | 0.35              |
> |                          | AS      | 0.49       | 0.48              |
> | MPGD                     | CP      | 0.56       | 0.45              |
> |                          | AS      | 0.48       | 0.35              |
> | SVDD                     | CP      | 0.82       | 0.82              |
> |                          | AS      | 0.59       | 0.57              |
> | CTRL                     | CP      | 0.94       | 0.94              |
> |                          | AS      | 0.93       | 0.93              |
>
> [1] Wu, Luhuan, et al. "Practical and asymptotically exact conditional sampling in diffusion models." Advances in Neural Information Processing Systems 36 (2024).
>
> [2] Phillips, Angus, et al. "Particle Denoising Diffusion Sampler." arXiv preprint arXiv:2402.06320 (2024).
>
> [3] Li, Xiner, et al. "Derivative-free guidance in continuous and discrete diffusion models with soft value-based decoding." arXiv preprint arXiv:2408.08252 (2024).
>
> [4] He, Yutong, et al. "Manifold preserving guided diffusion." arXiv preprint arXiv:2311.16424 (2023).

---

> ### Author Response · Authors · 2024-11-21
> **Continued Response**
>
> **Q. This paper does not perform an ablation analysis of the proposed method.**
>
> We have indeed conducted an ablation analysis on the guidance strength parameter $\gamma$, as detailed in Appendix H.3. Selecting an optimal KL weight is critical. In practice, we found that $\gamma$ values between 5 and 20 are generally effective, enabling accurate conditional generations for the compressibility task. Additionally, guidance strength $\gamma$ can be freely adjusted during inference (see Appendix D). Due to page limitations, we have included this ablation study in the appendix.
>
> ---
> **Weakness: How to formalize the problem into MDP, the state, the action, and the transfer function?**
>
> Thanks for reminding this! In the updated manuscript, we have included a detailed introduction to ​​the formulation of fine-tuning diffusion models via RL in Appendix B. If you find any part unclear or need any further clarification, please let us know.
>
> ---
>
> **Q: Why does the experimental verification compare mainly with methods without classifiers, despite CTRL being a splitter-based method?**
>
> Thank you for your question. While we are unsure what the reviewer specifically means by ***splitter-based method*** is  (and would appreciate further clarification), we understand the concern about comparisons with a classifier-based baseline. We would be happy to clarify more if we have a misunderstanding.
>
> We would like to emphasize that the traditional classifier-based diffusion models rely on mapping $x_t \to y$ (as noted in Table 1). A key drawback of this approach is the classifier’s accuracy in predicting $y$ from intermediate $x_t$​, which can lead to cumulative errors during the diffusion process. Reconstruction guidance methods have been proposed to address this. These methods approximate intermediate states $x_t$​ back to the original input space $x_0$​, allowing classifiers to be trained on $x_0 \to y$ directly.
>
> In the experimental part, we have included a classifier-based diffusion baseline as reconstruction guidance, providing a fair and comprehensive comparison.
>
> ---
>
> **Q. “The proposed method can be adapted to any off-the-shelf RL algorithms.”**
>
> Thank you for raising this concern. We agree with you. Our intention was simply to point out that many types of algorithms can be applied (though not all RL algorithms). We have revised our wording to “many types of off-the-shelf RL algorithms” accordingly.
>
> ---
>
> **Q. Citing the original paper proposing the PPO method for completeness.**
>
> Thank you for pointing this out. We have added a citation of the original PPO paper in the revised version.

---

> > ### Comment · Reviewer_Po57 · 2024-11-22
> > **Response to rebuttal**
> >
> > Thank you for the response. Through the author's responses, I have been able to clearly comprehend the method and the performance, and this aspect seems to have sufficiently addressed my inquiries.
> > Considering its potential contribution to the expansion of future research areas, I have decided to adjust the score upward.

---

> > > ### Author Response · Authors · 2024-11-23
> > >
> > > Thanks for reading our response and adjusting the score! We are happy that our clarifications help to address your concerns/inquiries. We are thankful for your constructive comments.

---

### Official Review · Reviewer_acmr · 2024-11-01

**Soundness:** 3
**Presentation:** 3
**Contribution:** 3
**Rating:** 8
**Confidence:** 4

**Summary:**

The paper introduces CTRL to enhance pre-trained diffusion models by integrating additional conditional controls through reinforcement learning. CTRL reformulates the conditional generation as an RL problem, where the reward function is derived from the conditional likelihood of labels given data, and the KL divergence from the pre-trained model acts as a regularizer. This method has key advantages over traditional guidance techniques, as it can use pairs of samples instead of triplets by leveraging conditional independence. CTRL is empirically validated on image generation tasks, demonstrating superior performance in single-task and multi-task settings, including cases requiring compositional controls, like generating images with specific compressibility and aesthetic properties.

**Strengths:**

1. The using of reinforcement learning to add conditional controls to diffusion models is novel. By reframing conditional generation as an RL problem, this approach leverages optimal policy learning to achieve conditional sampling without needing complex data dependencies. This use of RL significantly improves sample efficiency and offers a new, effective pathway for conditional generation, especially beneficial for complex, multi-condition tasks.
2. The comparison with prior work is comprehensive. The authors thoroughly contrast their RL-based CTRL method with traditional methods such as classifier guidance and classifier-free guidance, highlighting where CTRL overcomes limitations in sample efficiency, dataset requirements, and control precision. By detailing the theoretical and practical distinctions, such as the use of conditional independence to simplify dataset construction, the paper effectively demonstrates how CTRL builds upon and enhances past work, providing readers with a well-rounded understanding of its innovations.
3. Experimental results show that CTRL not only meets target conditions more accurately but also maintains high performance across diverse tasks with fewer data dependencies, underscoring its efficiency and robustness in practical applications.

**Weaknesses:**

1. While the core RL formulation and the connection to conditional diffusion are well-explained, the theoretical framework could be strengthened with a deeper exploration of the method's convergence properties and guarantees.
2. The experiments primarily compare CTRL with two standard methods: classifier guidance and classifier-free guidance. Including additional baseline methods, such as more recent variants of guided diffusion techniques or alternative conditional generation approaches, would provide a clearer picture of CTRL’s advantages and limitations.

**Questions:**

In the error analysis section, the paper briefly discusses three sources of error. Could these three types of errors be more precisely quantified?

---

> ### Author Response · Authors · 2024-11-21
> **Response**
>
> Thank you for the thoughtful and positive review! We’re glad you found our approach novel and appreciated our work’s contributions.
>
> In this response, we have: (a) agreed that further exploration of convergence properties and theoretical guarantees could strengthen the framework, which serves as a valuable direction for future work; (b) added more recent conditional methods as baselines according to your suggestion. **We have updated a new version of the manuscript with the changes highlighted in red.**
>
> ---
>
> **Q. Could these three types of errors be more precisely quantified?**
>
> In Section 4.3, we outline three potential sources of error that CTRL may encounter, primarily to set the stage for comparisons with existing methods in Section 5. Specifically, our goal is to demonstrate that CTRL achieves greater sample efficiency than classifier-free guidance in certain scenarios. We acknowledge the importance of providing more precise quantification of these errors and will pursue this as a focus in future research. Thank you again for your constructive insights!
>
> ---
>
> **Q. Including more recent conditional methods as baselines.**
>
> Thank you for raising this! We have included three additional baselines for both (1) the 4-class compressibility task (Section 6.1) and (2) the multi-task conditional generation (Section 6.2). Below, we briefly introduce the added baseline methods, all of which have used the same trained classifier adopted in CTRL for guidance:
>
> 1. **Sequential Monte Carlo (SMC)-based method**: Recent works [1,2] leverage resampling techniques in SMC to approximate distributions in diffusion models across a batch of samples (i.e., particle filtering). This method is training-free.
>
> 2. **SVDD** [3]: A decoding-based method that iteratively selects preferable samples during each diffusion step based on reward signals. For our conditioning tasks, the compressibility classifier and aesthetic score classifier serve as the reward functions. This method is also training-free and operates at inference time.
>
> 3. **MPGD** [4]: This method refines the predicted clean data $x_0$​ during inference based on a manifold hypothesis. While such refinement incurs additional computational costs during inference, it does not fine-tune diffusion model weights, remaining a training-free approach.
>
> We report the results below, showing that CTRL achieves good performance, outperforming all baselines in both tasks.
>  While SVDD and MPGD surpass SMC, they still fall short of CTRL. These results highlight CTRL’s superiority in achieving accurate and controllable generation, even compared to the most advanced conditioning methods.
>
> * 4-class compressibility task (Section 6.1)
>
> |                   | Accuracy ↑ | Macro F1 Score ↑ |
> |--------------------------|------------|-------------------|
> | Classifier-Free | 0.33       | 0.28              |
> | Reconstruction Guidance  | 0.45       | 0.45              |
> | SMC                      | 0.27       | 0.22              |
> | MPGD                     | 0.70       | 0.71              |
> | SVDD                     | 0.78       | 0.78              |
> | CTRL                     | 1.0        | 1.0               |
>
> * multi-task conditional generation (Section 6.2)
>
> |                   | Reward | Accuracy ↑ | Macro F1 Score ↑ |
> |--------------------------|---------|------------|-------------------|
> | Classifier-Free | CP      | 0.52       | 0.39              |
> |                          | AS      | 0.59       | 0.55              |
> | Reconstruction Guidance  | CP      | 0.61       | 0.55              |
> |                          | AS      | 0.66       | 0.62              |
> | SMC                      | CP      | 0.51       | 0.35              |
> |                          | AS      | 0.49       | 0.48              |
> | MPGD                     | CP      | 0.56       | 0.45              |
> |                          | AS      | 0.48       | 0.35              |
> | SVDD                     | CP      | 0.82       | 0.82              |
> |                          | AS      | 0.59       | 0.57              |
> | CTRL                     | CP      | 0.94       | 0.94              |
> |                          | AS      | 0.93       | 0.93              |
>
> [1] Wu, Luhuan, et al. "Practical and asymptotically exact conditional sampling in diffusion models." Advances in Neural Information Processing Systems 36 (2024).
>
> [2] Phillips, Angus, et al. "Particle Denoising Diffusion Sampler." arXiv preprint arXiv:2402.06320 (2024).
>
> [3] Li, Xiner, et al. "Derivative-free guidance in continuous and discrete diffusion models with soft value-based decoding." arXiv preprint arXiv:2408.08252 (2024).
>
> [4] He, Yutong, et al. "Manifold preserving guided diffusion." arXiv preprint arXiv:2311.16424 (2023).

---

> > ### Comment · Reviewer_acmr · 2024-11-26
> >
> > Thank you for your response. I believe this is an inspiring piece of work that deserves further exploration in the future.

---

> ### Comment · Area_Chair_HXvo · 2024-11-23
> **From AC.**
>
> Reviewer acmr: if possible, can you reply to the response?

---

### Official Review · Reviewer_ahRC · 2024-11-03

**Soundness:** 3
**Presentation:** 3
**Contribution:** 3
**Rating:** 6
**Confidence:** 3

**Summary:**

This paper proposes a novel method, CTRL, to address adding conditional control to diffusion models. CTRL consists of three main stages: constructing the augmented model with a pre-trained model, training a classifier as a reward model, and solving a reinforcement learning problem to introduce conditional control in the diffusion model. Compared to classifier-free guidance, CTRL is adaptable to more flexible offline datasets, and compared to classifier-guidance methods, CTRL avoids accumulated inaccuracies caused by predicting $y$ from $x_t$ at each denoising step. This makes CTRL applicable in a broader range of scenarios with a more controllable generation process. Finally, experimental results validate the effectiveness of CTRL in terms of controllability and generation quality in tasks involving compressibility and multi-task generation.

**Strengths:**

1. The problem the authors attempt to address—adding conditional control to pre-trained diffusion models—is highly valuable. The author’s use of RL modeling for this problem is novel, and the derivation process is both reasonable and rigorous.
2. Compared to previous methods, CTRL has more flexible requirements for offline datasets and does not require training a predictor $x_t\rightarrow y$, resulting in a broader range of applications and a more controllable generation process.
3. In the experimental section, compared to classifier-free guidance and classifier guidance, CTRL demonstrates superior performance in controllability for image generation.

**Weaknesses:**

1. The evaluation metrics used in the paper are insufficient; for instance, the author only verifies the controllability of CTRL without providing results on the image quality generated by CTRL.
2. The experimental tasks are limited. The author only uses compressibility and aesthetic pleasingness as conditional controls. It is necessary to compare more complex conditions, such as sketch, normal map, as suggested in [1].

[1] Adding Conditional Control to Text-to-Image Diffusion Models. ICCV, 2023.

**Questions:**

See weaknesses.

---

> ### Author Response · Authors · 2024-11-21
> **Response**
>
> We are deeply thankful for your recognition of our work’s significance and performance. In this response, we have provided additional evaluation metrics to illustrate the image quality generated by CTRL. We also provide a rationale for our experimental setup.  **Please note that we have updated a new version of the manuscript to incorporate changes, which are highlighted in red.**
>
> ---
> **Q. Evaluation metrics are insufficient; for instance, the author only verifies the controllability of CTRL without providing results on the image quality generated by CTRL.**
>
> Thank you for pointing this out! To address this, we provide additional evaluation metrics for CTRL and baseline methods, specifically BRISQUE [1] (lower values indicate better image quality) and CLIP score [2] (higher values reflect better text-image alignment).
>
> As shown, CTRL achieves better image quality than Classifier-Free Guidance and Reconstruction Guidance while achieving the highest alignment score.
>
> | Method                   | BRISQUE ↓      | CLIPScore ↑ |
> |--------------------------|----------------|-----------------------------------------------|
> | CTRL                     | 30.8 ± 1.5    | 26.8 ± 0.2                                   |
> | Classifier-Free Guidance | 33.2 ± 14.5   | 25.7 ± 1.0                                   |
> | Reconstruction Guidance  | 90.2 ± 10.1   | 22.8 ± 0.6                                   |
>
>
> [1] Mittal, A., Moorthy, A. K., & Bovik, A. C. (2012). No-reference image quality assessment in the spatial domain. IEEE Transactions on image processing, 21(12), 4695-4708.
>
> [2] https://lightning.ai/docs/torchmetrics/stable/multimodal/clip_score.html
>
> ---
>
> **Q.  The author only uses compressibility and aesthetic pleasingness as conditional controls.**
>
> We appreciate your suggestion to explore more complex conditions, such as those used in ControlNet. While these are indeed compelling directions, these works focus on introducing new neural network architectures with an existing guidance method.
>
> However, the focus of this work is developing a new principled approach to add conditional guidance with strong theoretical guarantees. Therefore, our current experimental tasks—compressibility and aesthetic pleasingness—were carefully chosen to validate the fundamental aspects of CTRL. Specifically, our contributions are as follows:
>
> 1. ***Framing Conditional Generation as an RL Problem***: We first frame conditional generation as an RL problem, providing a novel perspective. As noted by Reviewers ahRC and acmr, this framing is innovative and represents a significant advancement in guided/conditional generation.  We also formally prove that executing the soft-optimal policy in this RL would enable sampling from the target conditional distribution during inference. This finding provides the solid grounding of CTRL, as pointed out by the Reviewer FPqT.
>
> 2. ***Introducing the CTRL Algorithm***:  We propose the CTRL algorithm, which enjoys certain benefits over existing guidance methods. Specifically, CTRL eliminates the need to learn classifiers at multiple noise scales, as required in standard classifier guidance, and avoids fundamental approximations made by certain variants to bypass these challenges. In Section 5.2, we also highlighted CTRL’s distinct advantage in leveraging conditional independence, which leads to two benefits: (a) more sample-efficient training compared to classifier-free guidance and (b) could be applied for multi-task conditional generation task, which we argue is new in the literature as far as we are concerned.
>
> 3. ***Connecting CTRL to existing guidance methods theoretically***: We establish a close theoretical and practical connection between CTRL, classifier guidance, and classifier-free guidance, as pointed out by the Reviewer FPqT and acmr.
>
> Finally, we would like to note that our tasks with compressibility and aesthetic scores are inherently challenging. For instance, generating aesthetically pleasing images often conflicts with achieving high compressibility (Fig. 2(a)), making our second experiment—synthesizing images conditioned on these two scores—particularly demanding.
>
> We observe that CTRL effectively handles all 2 $\times$ 2 conditions (Fig. 2(c)) and outperforms baselines. This directly validates CTRL’s advantage of **leveraging conditional independence**, as outlined earlier.
>
> In summary, we think our experiments are sufficient to showcase the principled contributions of CTRL and its benefits over existing conditional methods.

---

> > ### Comment · Reviewer_ahRC · 2024-11-21
> >
> > Thank you for your detailed response, which has addressed most of my concerns. I would like to raise my score to 6. However, I still recommend that the authors test CTRL under more conditions, as this would better demonstrate the effectiveness of CTRL across various scenarios.

---

> > > ### Author Response · Authors · 2024-11-22
> > >
> > > Thanks for reading our response and raising the score! We appreciate your suggestion to explore more conditions for demonstrating further effectiveness. Thanks for providing the constructive feedback.

---

### Official Review · Reviewer_FPqT · 2024-11-03

**Soundness:** 3
**Presentation:** 3
**Contribution:** 2
**Rating:** 6
**Confidence:** 2

**Summary:**

This paper introduces an RL-based approach for integrating conditional control into pre-trained diffusion models. The primary contribution of the proposed algorithm is its RL objective, which is coupled with an analytical form of the KL divergence, allowing for seamless optimization using standard reinforcement learning algorithms. The authors benchmarked the proposed method against established controllable generation techniques, specifically classifier guidance and classifier-free guidance, and empirically demonstrated the effectiveness of the CTRL.

**Strengths:**

+ This paper is well-organized and the presentation is clear.

+ Unlike classifier guidance and classifier-free guidance, CTRL approaches controllable generation from a distinct perspective by framing conditional generation as a reinforcement learning problem and fine-tuning the diffusion model over the reverse diffusion process. While a few prior works (e.g., [1]) have explored similar perspectives, this work is distinguished by its use of a KL-regularized RL framework, deriving the analytical form of KL regularization through Girsanov’s theorem. Additionally, I especially appreciate that the authors included a discussion in the main text on the relationship between CTRL, classifier guidance, and classifier-free guidance, providing valuable context for understanding the nuances of these methods.

[1] Kevin Black, Michael Janner, Yilun Du, Ilya Kostrikov, Sergey Levine. Training Diffusion Models with Reinforcement Learning.

**Weaknesses:**

+ Theoretically, an RL formulation allows us to view equation (5) as a sequential decision-making problem, enabling the learning of value functions for each diffusion time step $s$ and policy optimization based on these estimated values. However, in this paper (Algorithm 1), the authors instead use the diffusion model to roll out a diffusion path, preserving the gradient of each intermediate $x_t$ (assuming this interpretation is correct), computing the loss, and finally backpropagating the gradient across the entire diffusion path. This approach likely demands higher GPU memory, as it requires gradient preservation throughout the generation process, as well as increased computation, given the need to backpropagate gradients through the model multiple times. These requirements could limit the scalability and broader applicability of the proposed method.

+ Besides, in order to achieve controllable generation, classfier guidance and classifier-free guidance need to train either the classifier or a finetuned model, while CTRL needs to train both of them. This complicates the overall procedure.

**Questions:**

+ While there are indeed numerous works that frame the reverse diffusion process as an MDP and use reinforcement learning to optimize the diffusion model, I am concerned that such optimization may lack sufficient constraints. To clarify, consider two consecutive time steps, $t$ and $t+1$ in the forward diffusion process, we have the following equation:
 $$p_{t+1}(x_{t+1})=\int p_t(x_t)\mathcal{N}(x_{t+1}; \sqrt{1-\beta_{t+1}}, \beta_{t+1}I)$$
Now, suppose we apply CTRL to fine-tune the reverse process. Since neural networks are used to approximate the score functions and are updated directly to maximize the objective without explicit constraints, it is likely that the above relationship will no longer hold after optimization. In other words, following reinforcement learning, the forward process might no longer correspond to the original stochastic differential equation (SDE). Could the authors provide insights into this concern? Specifically, will such a deviation impact the performance or generalization capability of the diffusion model?

---

> ### Author Response · Authors · 2024-11-21
> **Response**
>
> We are thankful for your appreciation of our work’s presentation and theoretical contribution. In this response, we have addressed your concerns by providing explanations on: (a) the implementation with Stable Diffusion, (b) the complexity of the training process, and (c) potential deviations from the pre-trained models.
>
> We are happy to answer more if further clarification is required.
>
> ---
>
> **Q. Backpropagating the gradients demands high GPU memory.**
>
>    Thank you for pointing this out. We utilized standard memory-efficient fine-tuning techniques for diffusion models, as described in [1] and [2]. Hence, memory efficiency is not a substantial practical bottleneck in our experiments. While the main paper does not go into these details due to space limitations, we extensively discuss the implementation details and computational costs in Appendix C and Appendix H.
>
> As outlined in Appendix C and following [1, 2], we employed two strategies to reduce the memory footprint of direct reward backpropagation:
>
>   1. Fine-tuning only the LoRA modules rather than the original diffusion model weights.
>   2. Utilizing gradient checkpointing to compute derivatives on the fly.
>
> These techniques enable backpropagation through all 50 diffusion steps of Stable Diffusion (using a DDIM scheduler) while adhering to hardware constraints. Specifically, these strategies make it possible to fine-tune Stable Diffusion with a batch size of 4, requiring approximately 30 GB of GPU memory.
> Additionally, we use gradient accumulation and multiple GPUs to scale the effective batch size to 128, enabling smoother optimization.
>
>
> In Appendix H.1 and H.2, we discussed further strategies to reduce computational cost and memory requirements.
>
>
> [1] Clark, Kevin, et al. "Directly fine-tuning diffusion models on differentiable rewards." ICLR 2024
> [2] Prabhudesai, Mihir, et al. "Aligning text-to-image diffusion models with reward backpropagation." ICLR 2024
>
> ---
>
> **Q: CTRL needs to train both classifier and diffusion models. This complicates the overall procedure.**
>
> Thank you for noticing this point. We acknowledge that our method requires training both a classifier and fine-tuning the diffusion model. However, we would like to highlight the unique advantages of CTRL:
>
> 1. **Inference Efficiency**: While classifier-based guidance avoids diffusion model fine-tuning, incorporating classifier gradients into the diffusion process significantly slows down inference. In contrast, CTRL retains the inference speed of pre-trained diffusion models, making the upfront fine-tuning costs worthwhile for achieving guidance-directed sampling with faster inference compared to classifier-based methods.
> 2. **Sample Efficiency**: As outlined in Table 1, classifier-free methods generally rely on large volumes of offline data for effective training. Although CTRL requires training a separate classifier, which classifier-free methods do not, this training process is more sample-efficient than diffusion model fine-tuning. Consequently, CTRL’s two-step approach leverages limited offline data more effectively than classifier-free methods.
>
> Finally, from a performance perspective, as validated in Sections 6.1 and 6.2, CTRL achieves more accurate samples than both baselines.
> In light of these considerations, we believe the two-step nature of CTRL does not complicate training but instead balances efficiency and performance.
>
> ---
>
> **Q: Following reinforcement learning, the forward process might no longer correspond to the original stochastic differential equation (SDE).**
>
> If we understand correctly, the reviewer is concerned about deviation from the pre-trained models induced by fine-tuning. We are happy to clarify more if the concern is different.
>
> Please note that fine-tuning, as applied in CTRL, naturally introduces deviations from the pre-trained model and its associated SDE, which is common in diffusion model alignment. Such deviation is inevitable as there is a mismatch between the pre-training data distribution and the target controllable distribution. However, if the deviation is significant, it may result in reward over-optimization.
>
> To address this, we follow [3] to enforce an explicit KL constraint to control such deviations. This approach ensures a balance between aligning with the pre-trained model and optimizing task-specific performance, making any deviations purposeful and beneficial rather than detrimental.
>
> As a general point, we note that in the context of diffusion model alignment and fine-tuning, many works have explored mitigation strategies for reward over-optimization [3,4,5].
>
>
> [3] https://arxiv.org/abs/2402.15194
>
> [4] https://arxiv.org/abs/2402.08552
>
> [5] https://arxiv.org/abs/2405.18881

---

> > ### Comment · Reviewer_FPqT · 2024-11-23
> >
> > Thanks for the reply, most of my concerns are addressed, and I have updated my score.

---

> > > ### Author Response · Authors · 2024-11-23
> > >
> > > Thanks for your time reading our responses and updating the score!

---

> ### Comment · Area_Chair_HXvo · 2024-11-23
> **From AC.**
>
> Reviewer FPqT: if possible, can you reply to the authors' response?

---

### Meta-Review · Area_Chair_HXvo · 2024-12-19

**Metareview:**

This paper outlines a way to fine-tune a diffusion model, allowing conditioning on additional information. The proposed method is pragmatic because it relies on standard implementations of fine-tuning, based on reinforcement learning.  The authors demonstrate that the proposed method is superior to classifier-free methods (at the additional cost of training a classifier).

The strengths of the paper are: (1) novel, distinct perspective on training conditional models; (2) good comparison with prior work; (3) availability of source code.

Because of those strengths, I lean towards acceptance.

I do have some residual concerns about presentation, which I hope will be addressed in the camera-ready version. Specifically, the abstract is very difficult to parse. For example, RL is mentioned in a way that suggests that classifiers and KL constraints are an integral part of it (in fact, both can be useful for RL in certain settings, but they are not an integral part of vanilla RL). Also, for some strange reason, the formalisation of the task as an MDP is deferred till the appendix.

**Additional Comments On Reviewer Discussion:**

Reviewers and AC agree that the paper has potential (see strengths in the meta-review)

The main reason I recommend rejection is because of  problems with presentation and baselines, as well as the fact that no reviewer was convinced enough to really champion the paper.

However, I wouldn't mind if this got accepted.

---------------------------------
Update: I have changed the meta-review to recommend acceptance.

---

### Decision · Program_Chairs · 2025-01-22

Accept (Poster)